# KidneyGenAfrica multi-cohort Genome-wide association study and polygenic prediction of kidney function in 110,000 Africans

Abram B. Kamiza [1,2,3,4,5,37], Tinashe Chikowore [6,7,8,37], Guanjie Chen [9], Oyesola Ojewunmi [1,2], Tafadzwa Machipisa [10,11], Feng Zhou [1,12], Richard Mayanja [1,13], Sounkou Toure [14], Opeyemi Soremekun [1,15], Christopher Kintu [1,2,16], Mariam Nakabuye [1,16,17], Mine Koprulu [2], Allan Kalungi [1,2], Robert Kalyesubula [16], Babatunde Salako [18], Oyekanmi Nashiru [19], Manuel Corpas [20,21], Cassianne Robinson-Cohen [22], Nora Franceschini [23], Cristian Pattaro [24], Anna Köttgen [25], Dorothea Nitsch [4], Claudia Langenberg [2,26], Catherine Tcheandjieu [27,28], Moffat Nyirenda [1,4], Andrew P. Morris [29], Jennifer Asimit [12], Eleftheria Zeggini [15,30], Charles Rotimi [9], Michele Ramsay [5], Adebowale Adeyemo [9], June Fabian [31,32], Amelia C. Crampin [3,33,34,35], Jean-Tristan Brandenburg [5,36] & Segun Fatumo [1,2,4] ✉

Kidney disease disproportionately affects populations of African ancestry, yet most genetic studies have focused on Europeans. Here, we present a three-stage genome-wide association study meta-analysis of estimated glomerular filtration rate in ~26,000 individuals across Eastern, Western, and Southern Africa and ~81,000 African-ancestry individuals in the diaspora. Continental African meta-analysis identifies four independent genome-wide significant loci, including two previously unreported loci. Pan-African meta-analysis identifies 19 independent loci, including three previously unreported loci. Fine-mapping reveals four loci with high causality probability, and phenome-wide analyses demonstrate pleiotropic effects on cardiometabolic and immunological traits. Notably, *APOL1* high-risk variants strongly associated with kidney disease in African Americans show markedly lower frequency and attenuated effects in continental Africa, indicating potential distinct genetic architectures. Polygenic scores from genetically similar populations significantly outperformed those from distant cohorts. These findings demonstrate the necessity of conducting genomic research across diverse African populations to enable equitable health outcomes.

As many as 850 million people worldwide are estimated to have kidney disease[1]. Most of these individuals live in low- and middle-income countries and experience disproportionate disability and mortality. Chronic kidney disease (CKD) is the third fastest growing cause of death globally and is predicted to be the fifth leading cause of years of life lost by 2040[2]. Prior studies in the US demonstrated that African Americans (West African ancestry) have a three-fold higher risk of developing kidney failure than other population groups[3]. In African

Americans, part of this risk is heritable, arising from high-risk kidney disease variants of the apolipoprotein L1 (*APOL1*) protein, which confer a survival advantage by protecting against *trypanosomiasis*. The impact of *APOL1* high-risk variants and other (genetic) factors on CKD in continental African populations remains understudied. From the available data, CKD risk is complex and heterogeneous, differing by region and within regions, with a younger age of onset by 5 to 20 years[4], reflecting the impacts of infectious and non-communicable diseases, food and water insecurity, occupational environmental exposures, heat stress, and climate change[5].

To assess kidney function, endogenous biomarkers such as creatinine or cystatin C have been used to estimate the glomerular filtration rate (eGFR)[6]. Genome-wide association studies (GWAS) have identified hundreds of genetic loci associated with eGFR[7-12], which are now being used to identify individuals at high risk of CKD using polygenic scores (PGSs). However, most genetic factors associated with eGFR have been identified in cohorts of European and Asian ancestry[7-10]. The only GWAS for eGFR in continental Africa was conducted in Uganda[12] and lacked the statistical power to detect significant loci due to a small sample size. Hence, a concerted effort to collect adequately powered and well-phenotyped studies with genetic data across diverse African populations is needed to identify the genetic loci associated with eGFR in individuals of continental African ancestry. This will help improve our understanding of the genetic factors influencing kidney function in this population and improve PGS predictions.

To address these knowledge gaps, we established KidneyGenAfrica[13], a collaborative partnership aimed at advancing research and training excellence in kidney disease genomics in Africa. This initiative unites independent research centers focused on kidney function genomics and capacity-building efforts to create a robust framework for large-scale genomic studies. The present study aimed to identify genetic loci associated with eGFR. We conducted GWASs for eGFR in each contributing cohort or study of individuals from continental Africa. We then conducted a three-stage regional meta-analysis using GWAS summary data from the Eastern, Western, and Southern African geographical regions. We also performed fine mapping, colocalization, functional annotation, pathway analysis, and phenome-wide association studies (PheWAS). We then tested the association between *APOL1* haplotypes and eGFR and assessed the performance and transferability of the PGSs.

## Results

### Continental Africa GWAS

We used GWAS data from individuals of African ancestry. In Eastern Africa, we included data from the Africa Wits-INDEPTH partnership for Genomic Studies [AWI-Gen] Kenya, Uganda Genome Resources [UGR], and the Africa America Diabetes [AADM] study, Kenya; in Western Africa we included AWI-Gen data from Ghana and Burkina Faso, as well as AADM data from Ghana and Nigeria; and in Southern Africa we included data from the Malawi Epidemiology and Intervention Research Unit [MEIRU], AWI-Gen, South Africa, and the African Research on Kidney Disease [ARK] Study, (Supplementary Table 1, Fig. 1a). Our primary GWASs from participating cohorts and studies identified hundreds of SNPs significantly ($p < 5 \times 10^{-08}$) associated with eGFR across Africa. To identify independent loci, we clumped all significant SNPs 500 kb around the lead SNPs. We found three independent loci in MEIRU, two loci in UGR wave one, one locus in ARK, one locus in AWI-Gen South Africa, one locus in UGR wave 2, and one locus in AWI-Gen Western Africa (Supplementary Table 2). The Manhattan and quantile-quantile (QQ) plots of the contributing GWASs are shown in Supplementary Fig. 1.

### Meta-analysis

In the regional meta-analysis, we defined a locus to be independent if it met the following criteria: (i) it reached genome-wide significance ($p < 5 \times 10^{-08}$) in the regional meta-analysis and nominal significance ($p < 0.05$) in at least two of the contributing cohorts within that region, and (ii) it showed a consistent direction of effect across all contributing cohorts in the region. Based on these criteria, we identified four independent loci associated with eGFR at the genome-wide significance level ($p < 5 \times 10^{-08}$) after clumping 500 kb around the lead SNPs (Table 1). Of the four independent loci, three were from the Eastern African meta-analysis and one was from the Southern African meta-analysis. The Manhattan and QQ plots for the regional meta-analyses are shown in Figs. 2a and 2b, respectively. The inflation factors for the regional meta-analyses were 1.013, 1.004, and 1.039 for the west, south, and east African meta-analyses, respectively, suggesting no obvious genomic inflation in our analyses. Of the four independent loci, rs4243063 (*LOC645752*) and rs73788952 (*OPRM1*) were novel (Table 1) and had not been previously reported to be associated with eGFR or CKD. Regional plots for the rs73788952 and rs4243063 loci are shown in Figs. 2c and 2d, respectively. Notably, independent loci

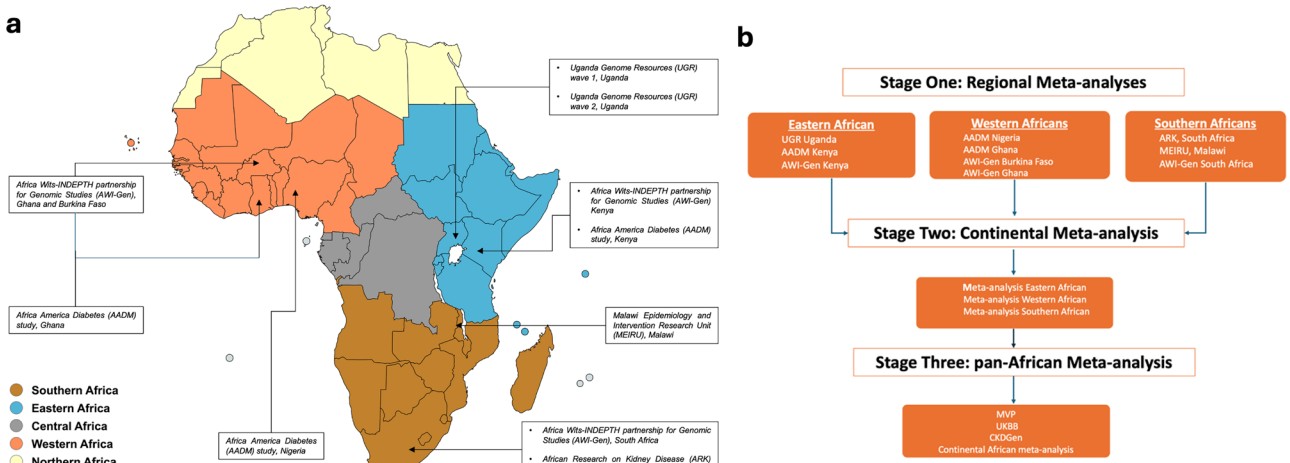

**Fig. 1 | Data sources and overview of the study design and approach. a** Map of Africa showing geographical regions included in the KidneyGenAfrican genome-wide association study meta-analysis. ARK South Africa (*n* = 1060), AWI-Gen-South Africa (*n* = 4527), MEIRU (*n* = 6380), UGR (*n* = 6407), AWI-Gen-Kenya (*n* = 1704), AADM-Kenya (*n* = 2069), AWI-Gen-Ghana and Burkina Faso (*n* = 3475), AADM-Ghana (*n* = 1060), AADM-Nigeria (*n* = 2069). **b** Flowchart showing three-stage meta-analyses performed to account for genetic diversity and population structure across Africa.

**Table 1 | Independent loci associated with eGFR in individuals of African ancestry in the regional meta-analysis**

| SNP | Genes | CHR | BP | EA | NEA | BETA | SE | P-value | Region | EAF-AFR' | EAF-EUR' | EAF-EAS' |
|---|---|---|---|---|---|---|---|---|---|---|---|---|
| rs6670659 | HSD3BP3 | 1 | 120090766 | C | G | 0.11 | 0.019 | 1.24E-08 | Eastern | 0.769 | 0.00 | 0.00 |
| rs73788952* | OPRM1 | 6 | 154342741 | G | A | 0.132 | 0.022 | 5.04E-09 | Southern | 0.092 | 0.00 | 0.00 |
| rs1706775 | SLC28A2 | 15 | 45586652 | T | C | 0.133 | 0.015 | 3.85E-17 | Eastern | 0.572 | 0.00 | 0.00 |
| rs4243062* | LOC645752 | 15 | 78189084 | T | C | -0.101 | 0.018 | 3.19E-08 | Eastern | 0.299 | 0.06 | 0.13 |

SNP single nucleotide polymorphism, CHR chromosome, BP base pair position, EA effect allele, NEA non-effect allele, EAF effect allele frequency, SE standard error, * previously unreported loci, AFR African, EUR European, EAS East Asian ancestry, 'effect allele frequency from this meta-analysis, 'effect allele frequency from 1000 Genome Project data, P-values are two-tailed calculated using GWAMA and adjusted for multiple comparisons.

associated with eGFR were not replicated in the other regional meta-analyses (Fig. 2e, Supplementary Data 1). We investigated this further and found that these loci were monomorphic in individuals of European and Asian ancestry (Table 1). However, these loci were replicated in contributing studies or cohorts within the geographical region of origin (Supplementary Data 2), with effect sizes in the same direction (Supplementary Fig. 2). Moreover, the Q and $I^2$ statistics of these loci were <3.2 and 0.446, respectively, except for rs73788952, which had a Q statistic of 8.001 and $I^2$ statistic of 0.750 (Supplementary Table 3), suggesting significant heterogeneity.

Our continental African meta-analysis identified one independent locus, rs1145085 (GATM), associated with eGFR after clumping all significant SNPs 500 kb around the lead variant. This locus has been reported to be associated with eGFR and CKD-related risk factors. In our phase three pan-African meta-analysis using data from continental Africa, MVP, UKB, and CKDGen consortium of individuals of African ancestry, we identified 19 independent loci associated with eGFR after clumping 500 kb around the lead SNPs (Table 2). Manhattan and QQ plots for the continental and pan-African ancestry meta-analyses are shown in Fig. 2a and 2b, respectively. Of the loci associated with eGFR in the pan-African ancestry meta-analysis, three had not previously been reported to be associated with eGFR or CKD. These loci included rs141647693 (ARG1), rs12595073 (SORD2P), and rs1918516 (SQRDL) (Supplementary Fig. 3). However, we noted that rs12595073 and rs1918516 were located upstream and downstream of GATM, respectively. To validate the independence of these loci, we conditioned the signal of rs12595073 on rs1145085 and rs1918516. Conditional analyses were performed using the --cojo-cond option in the GCTA software (v1.94.4)[14] with the AWI-Gen dataset as the genomic reference panel. The analysis parameters included a frequency difference threshold of 0.4 and a collinearity threshold of 0.9. Summary statistics were used, and genomic positions were extracted from a 5 Mb region surrounding GATM, including lead SNPs rs12595073, rs1918516, and rs141647693. Two independent conditional analyses were conducted: (i) rs12595073 was conditioned on rs1145085 and rs1918516, and (ii) rs1918516 was conditioned on rs1145085 and rs12595073. Our results showed no substantial change in significance (pC = 8.32e−08 compared to p = 2.69e−08). Furthermore, when we conditioned the signal of rs1918516 on rs12595073 and rs1145085, its significance was slightly reduced (p = 7.85e−09 vs. pC = 7.68e−07), suggesting that these loci are independent of each other, despite being located upstream and downstream of GATM.

### Fine mapping
Two of the regions were not suitable for fine-mapping as after removing all variants that were not present in at least 60% of the individuals, either the lead variant was removed (rs200950799) and all remaining variants had $p > 1.5 \times 10^{-5}$ or the lead variant (rs2096858) and a variant in high LD with it were the only significant variants with all others having $p > 1 \times 10^{-5}$. The overlapping regions were merged, resulting in 16 regions, to which we applied JAM as implemented in JAMdynamic, allowing for multiple causal variants. We highlight six regions where we identified a high-confidence variant with a PP of causality > 0.99 (Table 3 and Supplementary Data 3). For example, we prioritised missense variants rs334 in HBB and rs624307 in SLC25A45, each with a marginal posterior probability of causality (MPP) of 1.0. The variant rs334 is causal for the sickle cell trait and was previously found to be associated with eGFR, CKD, and kidney failure in African, African American, and US Hispanic/Latino populations[12,15–17].

### Colocalization signal recapitulates known eGFR-related genes
Three loci × GWAS × gene expression trait combinations showed strong evidence of colocalization with renal expression phenotypes, including SHROOM3 (Tubulointerstitial NepthQTL2, 4:76910318-77910318, H4PP 0.97), DUSP11 (Glomerulus NepthQTL2, 2:73464631-

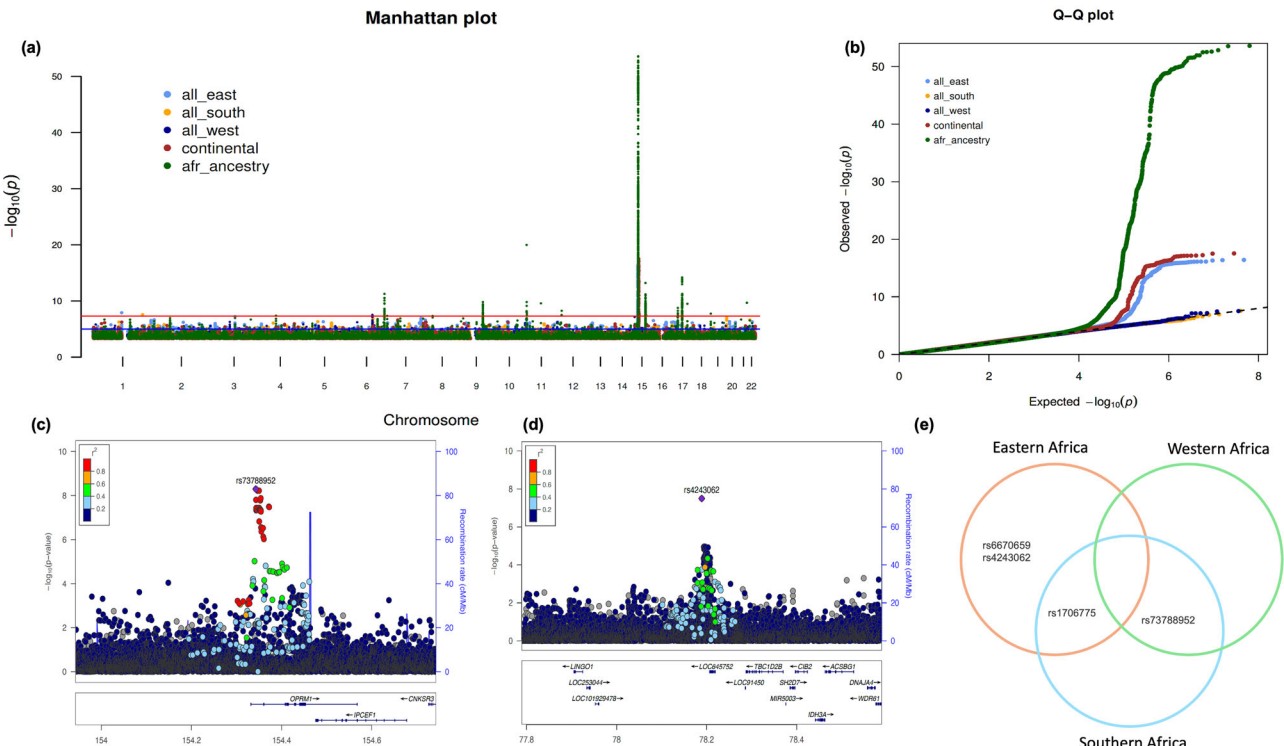

**Fig. 2 | Genome-wide association study of eGFR in individuals of African ancestry. a** Manhattan plots for East African (*N* = 10,481), South African (*N* = 11,967), West African (*N* = 6604), continental African (*N* = 29,052) and pan-African (*N* = 109,717) meta-analyses. X-axis represents the genomic position, the Y-axis represents the log10 of association *p*-values and the point above the dotted line represented variants significant at *p*-value < 5 ×10$^{-08}$ *P*-values are two-tailed calculated using GWAMA and METAL for eGFR meta-analyses, not adjusted for

multiple testing. **b** QQ plots for East African (*N* = 10,481), South African (*N* = 11967), West African (*N* = 6604), continental African (*N* = 29,052) and pan-African (*N* = 109,717) meta-analyses. X-axis represent the expected *p*-values and Y-axis represent the observed *p*-values. **c** Locuszoom plot showing associations around the OPRM1 regions. **d** Locuszoom plot showing associations around the LOC645752 region. **e** Venn diagram showing the number of independent genetic loci associated with eGFR identified by eastern and southern meta-analyses.

74464631, H4 PP4 0.85), and NAT8B (Kidney Cortex GTExV8, 2:73464631-74464631, H4 PP 0.91) (Supplementary Data 4). We also found 686 loci × GWAS × gene expression trait combinations with an H3 PP > 80%. Using a relaxed default prior and posterior probability of (PP.H4) ≥ 0.6 to assume shared genetics, we. We found no evidence of a shared eGFR risk variant with UPD pQTLs (Supplementary Data 5). This could be due to the small number of loci (20) analyzed and the fact that the OLINK assay does not probe the complete proteome.

## Gene-set, tissue-specific gene expression and pathway analysis

Our results showed that the mapped genes for the pan-African meta-analysis were highly expressed in the kidney (cortex and medulla) and vagina, whereas the continental African meta-analysis was not significantly enriched in any of these tissues (Supplementary Fig. 4) and (Supplementary Data 4). In the gene-based genome-wide analysis conducted across all African datasets, 17 genes achieved Bonferroni correction for multiple tests. However, only *GATM* and *SPATA5L1* were significant in a continental African meta-analysis. Notably, these two genes were also present in all other meta-analysis datasets (Supplementary Data 5). We prioritized 58 genes based on gene ontology terms obtained from MsigDB (Supplementary Data 6). Our findings revealed a strong enrichment of genes involved in various biological functions and pathways, such as hydrogen peroxide metabolic process, thyroid hormone regulation, detoxification, reactive oxygen species process, olfactory signalling pathway, and haemoglobin and haptoglobin binding across all African datasets (Supplementary Fig. 4 and Supplementary Data 7). Although we identified genes (*GATM, SPATA5L1, SLC47A1*, and *NFATC1*) that have been previously reported to be associated with creatinine, eGFR creatinine-based eGFR, urinary

albumin-to-creatinine ratio, and CKD, we also observed that some of the genes (*HBB, HBD, HBG2, HBE1, NRG1, and MMP26*) were associated with sickle cell disease, red blood cell traits, and malaria (Supplementary Fig. 4 and Supplementary Data 8).

## PheWAS

We found that 16 loci were linked to other metabolic traits, including body mass index (BMI), diabetes, CKD, and impedance measures. Nine loci were associated with cardiovascular traits, including key risk factors such as systolic blood pressure, hypertension, and ischaemic heart disease. Furthermore, 15 loci have been implicated in immunological traits, including various blood cell types. We found nine loci associated with psychiatric traits, including conditions such as depression and sleep-related behaviours, while five loci were linked to environmental traits, particularly responses to medications like HMG CoA reductase inhibitors and diuretics. Moreover, three loci were associated with dermatological traits, including skin tanning ability and skin colour, and four loci were associated with dietary-related traits, including the intake of processed meats, cereals, tea, and salt (Supplementary Table 4).

## APOL1 haplotypes distribution and association with low eGFR

The G1 allele frequencies were 6.8%, 9.9% and 12.2% in Eastern, Southern, and Western Africa geographical regions, respectively, whereas the G2 allele frequencies were 7.7 % in Eastern Africa, 10% in Western Africa and 16.6 % in Southern Africa. Moreover, the frequency of *APOL1* high-risk genotypes was highest in Southern Africa (7.4%), followed by Western Africa (4.2%) and Eastern Africa (2.2%) (Supplementary Table 5). We investigated the association of low eGFR with

**Table. 2 | Independent loci associated with eGFR in individuals of African ancestry in the pan-African meta-analysis**

| SNP | Genes | Region | CHR | BP | NEA | EA | EAF | Zscore | P-value | EAF-EUR | EAF-EAS |
|---|---|---|---|---|---|---|---|---|---|---|---|
| rs11894953 | TPRKB | Upstream | 2 | 73964631 | C | T | 0.451 | −5.546 | 2.92E−08 | 0.69 | 0.68 |
| rs4859682 | SHROOM3 | Intronic | 4 | 77410318 | C | A | 0.077 | −5.499 | 3.81E−08 | 0.43 | 0.20 |
| rs141647693* | ARG1 | Intergenic | 6 | 131810450 | C | T | 0.152 | 5.708 | 1.14E−08 | 0.01 | 0.00 |
| rs316020 | SLC22A2 | Intronic | 6 | 160669081 | G | A | 0.183 | 6.763 | 1.35E−11 | 0.11 | 0.04 |
| rs13230509 | UNCX | Intergenic | 7 | 1286192 | G | C | 0.180 | −5.654 | 1.56E−08 | 0.68 | 0.31 |
| rs4236709 | NRG1 | Intronic | 8 | 32410110 | G | A | 0.328 | 5.538 | 3.07E−08 | 0.81 | 0.78 |
| rs7848018 | C9orf3 | Intronic | 9 | 97748906 | C | A | 0.483 | −6.125 | 9.08E−10 | 0.02 | 0.15 |
| rs334 | HBB | Exonic | 11 | 5248232 | T | A | 0.099 | −9.393 | 5.82E−21 | 0.00 | 0.00 |
| rs77127179 | CCKBR | Downstream | 11 | 6293717 | G | A | 0.025 | −5.551 | 2.85E−08 | 0.00 | 0.00 |
| rs624307 | SLC25A45 | Exonic | 11 | 65144075 | C | T | 0.101 | 6.519 | 7.07E−11 | 0.00 | 0.00 |
| rs12595073* | SORD2P | ncRNA_intronic | 15 | 45153405 | G | T | 0.102 | −5.56 | 2.69E−08 | 0.04 | 0.39 |
| rs1145085 | GATM | Intronic | 15 | 45657804 | G | A | 0.133 | 15.4 | 1.63E−53 | 0.72 | 0.18 |
| rs1918516* | SQRDL | ncRNA_intronic | 15 | 46161483 | G | T | 0.137 | 5.772 | 7.85E−09 | 0.71 | 0.96 |
| rs8034430 | UBE2Q2 | Intronic | 15 | 76169004 | G | A | 0.096 | −7.502 | 6.30E−14 | 0.01 | 0.00 |
| rs2453585 | SLC47A1 | Intronic | 17 | 19447612 | T | A | 0.139 | −6.225 | 4.82E−10 | 0.16 | 0.31 |
| rs7208487 | FBXL20 | Intronic | 17 | 37543449 | G | T | 0.341 | −8.000 | 1.25E−15 | 0.16 | 0.24 |
| rs9895661 | BCAS3 | ncRNA_intronic | 17 | 59456589 | C | T | 0.488 | 6.294 | 3.10E−10 | 0.19 | 0.54 |
| rs56376587 | NFATC1 | Intronic | 18 | 77160235 | C | A | 0.151 | 5.632 | 1.78E−08 | 0.53 | 0.33 |
| rs2096858 | RNU6-1150P-201 | Intergenic | 21 | 45417467 | T | C | 0.016 | 6.361 | 2.00E−10 | 0.00 | 0.27 |

SNP single nucleotide polymorphism, CHR chromosome, BP base pair position, EA effect allele, NEA non-effect allele, EAF Effect allele frequency, EUR European. EAS East East Asian ancestry, * previously unreported loci; P-values are two-tailed, calculated using METAL and adjusted for multiple comparisons.

**Table 3 | Genomic regions with variants with a high posterior probability of causality**

| Region | Nearest Gene(s) | Variants with PP > 0.90 | MAF | Most severe consequence | PP | CS99 size |
|---|---|---|---|---|---|---|
| 7:786192-1786192 | *UNCX, MICALL2* | rs13230509 | 0.30 | regulatory region variant | 0.97 | 2 |
| 11:4748232-5748232 | *HBB* | rs334 | 0.06 | missense variant | 1.00 | 1 |
| 11:64644075-65644075 | *SLC25A45* | rs624307 | 0.09 | missense variant | 1.00 | 1 |
| 15:75669004-76669004 | *UBE2Q2* | rs8034430 | 0.11 | Intron | 0.97 | 3 |
| 17:58956589- 59956589 | *BCAS3* | rs9895661 | 0.49 | noncoding transcript exon variant | 1.00 | 1 |
| 18:76660235-77660235 | *NFATC1* | rs56376587 | 0.19 | Intron | 1.00 | 1 |

*PP* posterior probability of causality, *MAF* minor allele frequency, CS99; 99% credible set size.

**Table 4 | Association of *APOL1* and Low eGFR in continental Africa**

| | east AFR | south AFR | west AFR | All# |
|---|---|---|---|---|
| *APOL1* alleles | | | | |
| 0 | 108/5651 (1.9) | 113/5473 (2.1) | 48/2082 (2.3) | 269/13206 (2.0) |
| 1 | 44/1905 (2.3) | 92/3730 (2.5) | 26/1245 (2.1) | 162/6880 (2.4) |
| 2 | 11/167 (6.6) | 19/715 (2.7) | 2/146 (1.4) | 32/1028 (3.1) |
| Models: OR (95% CI) | | | | |
| Additive | 1.65 (1.24–2.22) | 1.05 (0.85–1.30) | 0.84 (0.55–1.25) | 1.16 (0.99–1.36) |
| Dominant | 1.54 (1.09–2.18) | 1.70 (0.81–1.41) | 0.85 (0.52–1.36) | 1.46 (1.20–1.78) |
| Recessive | 4.38 (2.12–9.03) | 1.04 (0.64–1.71) | 0.59 (0.10–1.90) | 1.52 (0.48–4.76) * |

#The results show meta-analysis with meta in R with a fixed effect model, except for the recessive model that showed significant heterogeneity. *OR (95%CI) are from the random effect model. We excluded the ARK cohort from South Africa and ARK was not included in the models shown because the cohort had few participants ($n = 3$) with low eGFR and we could not perform the analysis.

*APOL1* under additive, dominant, and recessive inheritance models, adjusting for age, age-squared, and the first two PCs. In Eastern Africa, we found a significant association between the additive (OR = 1.65, 95% CI 1.24–2.22), dominant (OR = 1.54, 95% CI = 1.09–2.18) and recessive (OR = 4.38, 95%CI = 2.12–9.03) models. Our Southern African and West African cohorts (from Ghana, Navrongo, and Burkina Faso, Nanoro) have fewer cases of individuals with low eGFR and *APOL1* high-risk alleles, which could influence the statistical power to detect an association. This could further demonstrate the difference in effect size of *APOL1* risk alleles in a population-based study compared to a case-control study. In the overall cohorts, only the *APOL1* dominant model was significantly associated with low eGFR (OR = 1.46, 95% CI = 1.20–1.78) (Table 4).

### Polygenic scores
We found that PGSs derived from Southern Africa performed and predicted better in the MEIRU cohort than those derived from other regions, including the multi-ancestral meta-analysis (Fig. 3c). The predictive performance of PGSs were $R^2 = 0.11\%$ with $p$-value = 0.037 for Southern Africa, $R^2 = 0.00028\%$ with $p$-value = 0.916 for Eastern Africa, $R^2 = 0.022\%$ with $p$-value = 0.347 for Western Africa, $R^2 = 0.012\%$ with $p$-value = 0.491 for continental Africa, $R^2 = 0.022\%$ with $p$-value = 0.357 for pan-Africa, $R^2 = 0.068\%$ with $p$-value = 0.101 for Africa, and $R^2 = 0.007\%$ with $p$-value = 0.596 for the multi-ancestry meta-analysis (Fig. 3c). We also computed the PCA plot and found that the MEIRU and Southern Africa meta-analyses were genetically closer to each other (Fig. 3d), suggesting that genetic distance among base GWAS summary data, training, and validation cohorts are crucial in the development and application of PGSs.

### Discussion
The newly established KidneyGenAfrica Consortium has assembled summary statistics for eGFR GWAS from 10 continental African cohorts across three geographical regions in Africa, with up to 26,000 individuals. We identified two loci previously not reported to be associated with eGFR in a regional meta-analysis and three previously

unreported loci in the pan-African meta-analysis. Interestingly, we found several loci that have been previously reported to be associated with eGFR. Of the previous loci, the most well-established is rs1145085 (GATM). GATM plays a crucial role in creatine biosynthesis and has been reported to be associated with eGFR, creatinine levels, and CKD in individuals of European and Asian ancestry.

The unique genetic background of individuals of African ancestry provides an opportunity to identify eGFR-associated variants that are common in continental African populations but monomorphic or rare in other populations. Our analysis revealed that rs6670659 (HSD3BP3), rs73788952 (OPRM1), and rs1706775 (SLC28A2) were common genetic variants in Africa, yet monomorphic in European and Asian populations[18], underscoring the advantage of conducting genetic studies in individuals of African ancestry. We also found that our regional genetic loci were not replicated ($p < 0.05$) in region-based meta-analyses or in datasets such as UKB and MVP. African populations exhibit exceptionally high genetic diversity and faster, more heterogeneous LD decay than Europeans and East Asians, with greater variation within Africa than between Africans and Eurasians. This makes exact GWAS lead variants less likely to replicate uniformly across East, West, and Southern Africa even when the underlying causal loci are shared[19–23]. Regional LD patterns within Africa are not uniform, with documented differences among populations from West, East, and Southern Africa that affect tag SNP portability and imputation accuracy. A previous study reported divergent patterns of LD within Africa, including instances where alleles that were in positive association in one population were in negative association in another, and a resequencing study at *IL13* found divergent patterns of LD across West and East African populations[23], underscoring region-specific haplotype structures that can impede exact SNP replication across regions. Haplotype-based analyses further show substantial differences in LD extent and private haplotypes across sub-Saharan populations, directly impacting replication of index SNPs identified in one African cohort when tested in another[19,20,23,24]. However, our loci were replicated ($p < 0.05$) with the same effect direction in more than one of contributing cohorts or studies to the regional meta-analyses.

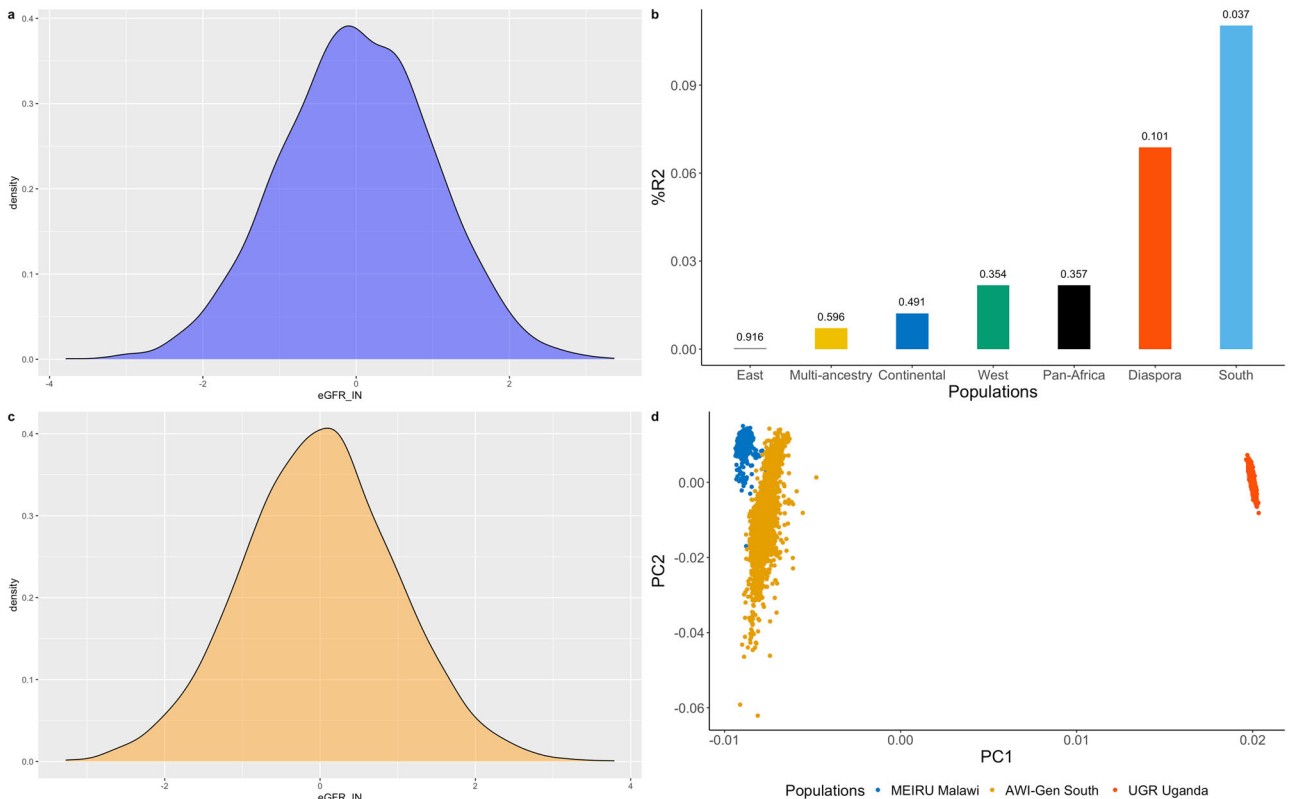

**Fig. 3 | Polygenic scores (PGSs) of eGFR in individuals of African ancestry.** The MEIRU cohort was randomly divided into testing and validation cohorts. **a** Distribution of eGFR in the PGSs testing ($n = 3180$) cohort (**b**) distribution of eGFR in the validation ($n = 3180$) cohorts. **c** R2 of PGSs in Southern, Eastern, Western, continental, pan-African, and multi-ancestry meta-analyses, with their respective *p*-values on top of the bar plot. **d** Principal component analysis showing the genetic distance among the continental African cohorts.

Notably, one of the previously unreported loci identified in the Southern African regional meta-analysis, rs73788952, reached genome-wide significance and showed a consistent effect direction across the contributing cohorts. However, this variant also displayed substantial statistical heterogeneity ($I^2 = 75\%$, Q $p = 0.018$), suggesting differences in the magnitude of association across studies. This may reflect local ancestry differences or region-specific environmental exposures that modify the genetic effect. Importantly, such heterogeneity is not uncommon in GWAS. For example, the UMOD–PDILT locus, one of the strongest and most well-replicated signals for eGFR and CKD in European populations, also shows high heterogeneity ($I^2 = 60\%$)[25]. In a recent trans-ethnic analysis using MR-MEGA, the heterogeneity at the UMOD locus was not attributable to ancestry differences ($p = 0.59$), highlighting that effect size variability can persist even at robust loci[26]. These findings underscore the need for continued investigation into region-specific genetic architecture and environmental modifiers of kidney function across diverse African populations.

Similarly, the unique LD pattern among individuals of African ancestry facilitated the identification of loci that drive eGFR and CKD through fine mapping analysis. We identified five loci (*HBB, SLC25A45, UBE2Q2, BCAS3, and NFATC1*) with a PP > 0.9. These loci have been previously associated with kidney phenotypes. Moreover, our colocalization recapitulated the known associations with a high H4 PP in different expression contexts for the same genes. The high number of H3 PP indicates the possibility of violation of the single causal allele assumption. This violation may have been due to mismatches between our samples and those included in the eQTL databases[27], which are skewed toward European ancestry.

Our PheWAS revealed biologically relevant associations beyond eGFR, highlighting the impact of lead variants from our pan-African meta-analyses on various metabolic traits, including CKD and related risk factors. We observed significant links to common comorbidities such as cardiovascular conditions (e.g., obesity, hypertension, diabetes) and immunological traits (e.g., red blood cell characteristics). Notably, we found associations with dermatological traits (ease of skin tanning and skin colour) and psychiatric traits (depression, sleep patterns, alcohol, and tobacco use)[28–34].

Given that many of our findings were previously reported primarily in non-African continental populations, further research is crucial for the African population and should focus on the traits identified in our PheWAS as potential early indicators of eGFR decline (e.g., obesity and hypertension) and investigate lifestyle factors such as poor dietary patterns. A recent study of over 8355 West Africans from Nigeria and Ghana found G1 frequencies of 31.6% and 25.8% for the two *APOL1* high-risk alleles in cases and controls, respectively[35]. Substantial heterogeneity in *APOL1* alleles (for example, G2 allele frequency within West Africa, for example, 6–16.7% among the Yoruba in Nigeria and 6–24.4% across West African populations) has been reported in previous studies, highlighting the influence of demographic structure on *APOL1* distribution[36,37]. In the present study, we observed a significant association with low eGFR in Eastern Africa, whereas other African populations studied did not show such an association. Albuminuria has been linked to *APOL1* high-risk genotypes in a previous study involving continental Africans. A recent study by Gbadegesin et al. reported an association between APOL1 variants and CKD in cohorts from Nigeria and Ghana[35]. Their definition of CKD included an eGFR <90 ml per minute per 1.73 m², a urinary albumin-to-creatinine ratio ≥30 mg/g, or both. In addition, a proportion of their cases included individuals with biopsy-confirmed glomerular disease and those with sickle cell disease, which is a known risk factor for kidney impairment. In contrast, our study used a cross-sectional, population-based approach that

relied on a single measurement of eGFR to define low eGFR. Although eGFR is a useful indicator of renal function at the population level, it is not the same as a clinical diagnosis of CKD, particularly in the absence of longitudinal data. Thus, our classification of kidney function status for *APOL1* analysis, the demography of participants and study design may contribute to the lower effect sizes observed in our study compared to previous findings[35,38]. Detection of *APOL1* with a stronger effect size is mostly observed in end-stage kidney disease, focal segmental glomerulosclerosis, and HIV-associated nephropathy[39].

We found that PGSs derived from Southern Africa performed and predicted better in the MEIRU cohort than those derived from other regions, including a multi-ancestry meta-analysis. Previous studies have indicated that PGSs derived from multi-ancestry summary data predict diverse ancestries better than single-ancestry PGSs[40,41]. Recent findings from the COGENT-Kidney consortium[42] found that polygenic scores derived from African Americans and Africans consistently outperformed European ancestry-specific scores for prediction into the African American test GWAS, despite the more than 10-fold difference in sample size[11]. However, all polygenic scores explained a low proportion of eGFR variance in West Africans from Nigeria and Ghana, supporting the findings of this study focus on African participants. Moreover, other studies have suggested that large discovery summary data enhances the performance and transferability of PGSs. However, this was not the case in the present study. After excluding MEIRU, data from Southern Africa contributed 4.14% ($n = 4527$) to the pan-African meta-analysis, yet it outperformed other PGSs, including diasporan ($n = 80,665$), pan-African ($n = 109,416$), and multi-ancestry ($n = 1,046,070$) meta-analyses. The poor performance of PGSs may be driven by differences in the genetic architecture of eGFR and possibly by the genetic distance between populations used for testing, validation, and sources of the summary data. We noted that the Southern populations in the Southern African meta-analysis clustered close to the MEIRU cohort in Malawi. Moreover, we found more previously unreported loci in the Southern African meta-analysis than in other regions. Notably, loci found to be associated with eGFR in our regional meta-analyses were not replicated in other regional meta-analyses, further suggesting that genetic distance is crucial for eGFR meta-analysis and PGSs development in continental Africa.

Our study had several strengths, including the identification of previously unreported loci in regional and pan-African meta-analyses. Our fine mapping identified loci that drive eGFR association in individuals of African ancestry and further identified eGFR-related variants and variants associated with CKD-related traits, metabolic traits, and immunological traits. However, our study was limited by the ancestry mismatch between our data and the GTEx data used for colocalization, the lack of replication in regional meta-analyses, varied eGFR transformation used in the pan-African meta-analysis, high genetic differences among continental African populations, and varied ascertainment procedures across the cohorts, which likely impacted the GWAS results by influencing sample representativeness, genetic diversity captured, and the observed genetic architecture of eGFR. For example, the MVP cohort drawn from clinical settings may be enriched for individuals with lower eGFR or kidney disease, potentially increasing the power for disease-related variants but overestimating effect sizes for the general population. Conversely, population-based cohorts would offer more generalized findings. Crucially, the diverse geographic origins of our African cohorts, alongside African American populations, meant sampling distinct ancestral genetic backgrounds, which may directly affected allele frequencies and the discovery of region-specific loci.

In conclusion, this is the largest GWAS of eGFR conducted in continental Africa, surpassing the sample size of the previous analysis in the region by eightfold. We identified regional and pan-African-specific loci associated with eGFR in individuals of African ancestry. PGSs derived from cohorts with a close genetic distance between discovery and validation cohorts performed better than PGSs derived

from other regional meta-analyses, including multi-ancestry summary statistics. Our results indicate substantial heterogeneity in the genetic architecture of eGFR and CKD across continental Africa because it is driven by different genetic loci across regions.

## Methods

### Data sources for KidneyGen Africa and comparative analyses

All datasets used in this analysis were approved by their respective institutional review board. We used GWAS data from individuals of African ancestry (Supplementary Table 1, Fig. 1a). The AWI-Gen study participants were drawn from six study sites in Burkina Faso, Ghana, Kenya, and South Africa[43], ensuring a balance of Western, Eastern, and Southern African populations from rural and urban settings. Of the individuals recruited in AWI-Gen, 3475, 2069, and 4527 were from Western, Eastern, and Southern Africa, respectively. AADM is an ongoing study investigating the genetic epidemiology of non-communicable diseases (NCDs) in Africa. The AADM study comprised 5231 participants recruited from the University Medical Centers in Accra and Kumasi in Ghana, Enugu, Ibadan, and Lagos in Nigeria, and Eldoret in Kenya[44]. UGR is a population-based cohort of 6407 individuals from 25 neighbouring villages of Kyamulibwa, in the countryside of southwest Uganda in East Africa[45]. MEIRU, previously called the Karonga Prevention Study (KPS), collected data from 18,000 individuals living in urban and rural communities in Lilongwe and Karonga, Malawi[46]. Of these individuals, 7000 with phenotypic data were genotyped using the H3A genotyping array. Million Veteran Program (MVP) is an ongoing cohort study designed to investigate the genetic influences on health and disease among veterans in the US. The MVP has more than 1,000,000 individuals, of which 650,000 have been genotyped, including over 120,000 participants of genetic similarity to African ancestry[47]. Summary statistics for eGFR GWAS of African ancestry in MVP were obtained from 57,000 participants[48]. UKB includes genetic and phenotypic data of nearly 500,000 individuals aged 40–69 years[49]. Of these individuals, approximately 7000 were of African ancestry. CKDGen is a consortium of multi-ancestry population-based studies[50] set up to uncover genetic loci associated with kidney function and disease in individuals of different ancestries. In this analysis, we included 16,473 individuals of African ancestry from CKDGen.

### Quality control and imputation

Sample and SNP quality controls (QC) were performed for all genotyped data from the participating cohorts and studies in continental Africa. We excluded individuals who were highly related to the identity by descent (IBD) PIHAT > 0.9, had discordant sex information between the genotyped and phenotype data, had high genotype missingness, and an outlying heterozygosity rate. We also excluded SNPs with a call rate <99%, deviating from the Hardy-Weinberg equilibrium (HWE) at p < 0.00005, with minor allele frequency (MAF) < 0.005, located on sex and mitochondrial chromosomes, and ambiguous and multi-allelic SNPs. Individuals and SNPs that passed quality control were used for imputation in the Sanger Imputation Server, using the African Genome Resources (AGR) imputation panel as a reference for imputation. Prephasing of all genotyped data was performed using EAGLE2 and imputation was performed using the PBWT algorithm. After imputation, we further removed SNPs with an info score<0.3, SNPs with a MAF < 0.005, those violating the HWE assumption, monomorphic SNPs, multiallelic SNPs, and palindromic SNPs. The final quality-controlled imputed datasets were used for downstream analysis. GWAS summary statistics data from the UKB, MVP, and CKDGen were quality-controlled using easyQC[51]. We then lifted all GWAS summary data to be on the same genome build using Crossmap[52].

### Phenotype definition and transformation

eGFR was calculated using the CKD-EPI equation without using the coefficient for race-based adjustment[53]. The distribution of eGFR

varied by geographical region (Supplementary Fig. 5a); therefore, we performed an inverse-rank normal transformation of eGFR (Supplementary Fig. 5b). Residuals of the linear model of eGFR on age, age[2], and sex were inverse-rank normal transformed and used as outcomes in the GWAS.

## Association analyses

For association testing, we used a linear mixed model implemented in GEMMA[54] using the dosage format of the genotype data. The genomic relationship matrices (GRMs) were calculated on the pruned dataset from each contributing cohort (filtered for MAF < 0.01, missingness <0.05, and pruned in Plink with the option --indep-pairwise 50 5 0.2). The GRM was included in the linear mixed model to adjust for the remaining relatedness among individuals of African ancestry. We also performed sex-stratified association analysis. The output from the GEMMA from each participating cohort was used for the meta-analysis to increase the power of discovering additional previously unreported loci.

## Meta-analysis

GWAS results from participating cohorts and studies were pooled into a three-stage meta-analysis (Fig. 1b): (1) regional meta-analyses using the fixed-effect method implemented in the GWAMA package[55]. We used GWAS summary data from Eastern Africa (Kenya-AWI-Gen, UGR, and Kenya-AADM), Western Africa (Ghana-AWI-Gen, Burkina Faso-AWI-Gen, Ghana-AADM, and Nigeria-AADM), and Southern Africa (South Africa-AWI-Gen, MEIRU, and ARK). (2) The results of the three (Eastern, Western, and Southern Africa meta-analyses) were pooled into a continental meta-analysis using methods implemented in Metasoft software[56] to assess heterogeneity among the three geographical regions. (3) Finally, we performed a pan-African meta-analysis using summary data from continental Africa, MVP, UKB, and CKDGen using Stouffer's method, implemented in METAL[57]. Notably, Stouffer's method only uses $p$-value and the direction of effect to perform a meta-analysis.

## Fine mapping

In the pan-African meta-analysis, we identified lead variants using clumping and then constructed 1 Mb regions centred at the lead variants; one region was larger than 1 Mb due to the merging of three overlapping regions. We included all biallelic variants with allele frequencies > 0.01, which were present in at least 60% of the individuals in the pan-African meta-analysis. In addition, we retained only variants that were present in the 1000 Genomes African super-population reference panel[58]. Alleles were flipped in the GWAS as necessary, so that the effect alleles matched the reference panel. As the Stouffer method of meta-analysis as implemented in METAL outputs the Z-statistic, $z = \hat{\beta}/SE$, rather than effect estimates $\hat{\beta}$ and their standard errors, we used an approximation[59], based on the trait being standardised to mean 0 and variance 1:

$$\hat{\beta} = z/\sqrt{2p(1-p)(n+z^2)} \text{ and } SE = 1/\sqrt{2p(1-p)(n+z^2)} \quad (1)$$

We performed fine mapping of the association signals in the pan-African meta-analysis using a multiple causal variant approach with a dynamically selected maximum number of causal variants (JAMdynamic in the *MGflashfm* R package[60]) based on the Joint Analysis of Marginal summary statistics (JAM)[61]. The multiple causal variant approach uses an extended version of JAM that infers joint LD-adjusted multi-SNP models, highlighting the best SNPs and combinations of SNPs using a Bayesian sparse regression framework[61]. This requires the effect size estimate $\hat{\beta}$ for each variant, the effect allele frequency, and an estimate of the SNP correlation matrix, which we estimated from the 1000 Genomes African reference panel[58]. We set jam.nM.iter = 5 (ie running 5 million iterations in JAM stochastic search). The maximum

number of causal variants is dynamically set based on the data, with the aim of achieving a parsimonious model using the JAMdynamic function[60] which also outputs 99% credible set sizes (CS99) and SNP groups. CS99 was constructed by sorting the posterior probabilities (PPs) of the multi-SNP models in decreasing order and selecting the minimum number of SNPs models, such that their cumulative PP was ≥ 0.99. SNP groups were constructed such that variants in the same group had an LD r2 > 0.8, rarely appeared in a model together, and had a variant PP of inclusion > 0.01. These SNP groups help to refine credible sets. We used Locuszoom stand-alone v1.4 for regional association plots based on the AFR LD database[62].

## Colocalization

To identify putative molecular traits mediating effects of the GWAS-identified variants in significant loci, we tested 53 expression contexts (49 GTEx V8 tissues, 2 tissues from NepthQTL2 and 1000 Genomes cell lines expression trait from MAGE V0.1). Summary association data from GTExV8, NepthQTL2, and MAGE v.0.1 were downloaded and merged with GWAS results within the significant loci. Bayesian colocalization analysis was performed in the 1 Mb region around the index significant SNPs. For GTExV8 and MAGEv.0.1, harmonization included lift-over of GWAS data to the hg38 build and removal of duplicate sites. NephQTL2 glomerulus and tubulointerstitial cis eQTL summary data were downloaded, indexed, and merged with GWAS data in 1 Mb regions centered on the lead variants. Lifting-over and interval retrieval of tabix-indexed summary data operations were performed using the *Bigsnpr*[63] and *Seqminer* R packages[64]. Evidence of shared causal variants was assessed using the *coloc.abf* function in the *coloc* R package[65]. *Coloc*. abf outputs the PPs of five hypotheses: null of no causal allele within a locus (H0 PP), causal allele for only one target trait (H1 or H2), two distinct causal alleles (H3 PP), and posterior probability of a shared causal allele between the two target traits (H4 PP). The method was applied with default priors for GWAS × tissue expression × gene × locus combinations, and an H4 PP > 80% was set to indicate strong evidence of correlated signals. We also performed a Bayesian-based colocalization analysis with the coloc.fast function (https://github.com/tobyjohnson/gtx) between the independent loci and protein quantitative trait loci (pQTL) using Ugandan proteomic data (UPD)[66].

## Gene-set, tissue-specific gene expression and pathway analysis

To uncover SNP associations at the gene level and understand how genes are linked to biological pathways involved with eGFR levels, we used FUMA's gene-based test and gene-property analysis tool MAGMA[67]. The gene-based P-value was computed for protein-coding genes by mapping SNPs to 19306 protein-coding genes in the pan-African meta-analysis ($0.05/19306 = 2.59 \times 10^{-6}$) and ($0.05/19034 = 2.627 \times 10^{-6}$) for the continental African meta-analysis. Gene property analyses were conducted using gene-based tests, which allowed us to assess tissue-specific gene expression profiles and associations with disease-related genes in 53 tissue types using GTEx v8[59]. The Bonferroni correction was employed to account for multiple tests. LD across SNPs and genes was calculated using the 1000 Genome Phase 3 African reference panel. Prioritized genes were further investigated for overrepresentation in various gene sets using the GENE2FUNCTION tool in FUMA to identify biological processes associated with eGFR creatinine-based eGFR from the Molecular Signatures Database (MsigDB)[67]. We also investigated the extent of overlap between the eGFR association signals and those reported in previous GWAS studies listed in the GWAS Catalogue. Enrichment $p$-values were used to determine the proportion of overlap in the genes.

## PheWAS

We conducted a Phenome-Wide Association Study (PheWAS) to investigate the relationships between LD-pruned genome-wide

significant (GWS) eGFR SNPs ($p < 5 \times 10^{-08}$) and various human phenotypes, utilizing data from the GWASATLAS database. We input the rsID numbers of our lead SNPs into the GWASATLAS interface[68], which includes SNPs with $p < 0.05$. This platform standardizes SNPs at identical genomic coordinates for a comprehensive analysis. After identifying SNP-phenotype associations, we applied a multiple hypothesis testing correction, adjusting the significance threshold to $p < 0.05$, divided by the number of tests, to ensure robust statistical reliability.

## APOL1 variants and haplotypes analysis

We investigated the distribution of *APOL1* variants in Eastern Africa (AWI-Gen-Kenya and UGR), Southern Africa (AWI-Gen-South Africa, MEIRU, and ARK cohorts), and Western Africa (AWI-Gen-Ghana and AWI-Gen-Burkina Faso). The following imputed SNPs were extracted for the cohorts to define *APOL1* haplotypes: presence of both rs73885319 (p.S342G) and rs60910145 (p.I384M) as G1 and rs12106505 (proxy SNP for Indel rs71785313: p.N388Y389/-) for G2. For the MEIRU cohort, genotyped data were used for rs60910145. Individuals with high-risk haplotypes carry two risk alleles: homozygous G1/G1, homozygous G2/G2, or compound heterozygous G1/G2. We categorized others as low risk (G0/G0, G0/G1, and G0/G2). We performed an association analysis of low eGFR (<60 ml/min per 1.73 m2) with *APOL1* under additive, dominant, and recessive models using logistic regression and adjusted for age, age-squared, and the first two genetic principal components (PCs). All analyses were carried out in R. We could not perform logistic regression for the ARK cohort because of the small number of participants ($n = 3$) with a low eGFR. We then performed a meta-analysis of the odds ratio (95% confidence interval (CI)) using the *meta* package in R[69] for cohorts to obtain single results for Eastern, Southern, and Western Africa. We reported fixed effects results as OR (95% CI) for the *APOL1* meta-analysis; otherwise, random effects results when the heterogeneity test *p*-value across the cohorts was significant.

## Polygenic scores

We then developed PGSs using PRSice-2[70] in the MEIRU cohort ($n = 6380$), which were divided into testing ($n = 3190$) and validation ($n = 3190$) cohorts. GWAS summary data from regional (east, west and south), continental, diaspora (MVP + UKB+CKDgen), pan-African, and multi-ancestry meta-analyses were used to derive PGSs in the MEIRU testing cohort. We then compared the performance and transferability of the PGSs in the MEIRU validation cohort. To avoid inflation of PGSs caused by sample overlap, we excluded the MEIRU dataset from all GWAS summary statistics data that were used to derive PGSs. For multi-ancestry, we used summary data from a previous study with more than 765,348 individuals[7]. Of these individuals, 567,460 were Europeans, 165,726 were East Asians, 13,842 were African Americans, 13,359 were South Asians, and 4961 were Hispanic. For PGS construction, SNPs from the meta-analyses were clumped based on LD. We further clumped SNPs at different distances and r2 thresholds to determine the optimal model for PGSs construction and replication (Supplementary Data 9). We also tested the optimal p-value threshold for selecting clumped SNPs for inclusion in the final PGS. The p-value threshold, which accounted for the highest variance of trait R2, was selected as the best PGS construction. PGSs were calculated by multiplying the weight of the SNPs by the number of risk alleles (0/1/2) carried by each individual using the algorithm implemented in PRSice-2 software[70]. The generated PGSs were incorporated into a linear regression model to explain eGFR. In the null model, we included age, sex, and five principal components. For the full model, we included PGSs, sex, age, and five principal components. R2 is the difference between the fully adjusted and null models (PGS-R2 = full model R2 – null model R2). PCA was performed after removing related individuals with pairwise IBD PIHAT > 0.185 and pruning highly correlated SNPs. The PCA was calculated using PLINK and visualized using the ggplot2 package in R.

## Reporting summary

Further information on research design is available in the Nature Portfolio Reporting Summary linked to this article.

## Data availability

Full summary statistics relating to the Million Veteran Program (MVP) studies are available at dbGAP accession phs001672.v2.p1. Individual-level data, phenotype, and genotype data of the continental Africa cohort are available to researchers under managed access on European Genome-phenome Archive (EGA): UGR: EGAS00001001558, EGAD00010000965, AWI-GEN: EGAD00001004448, and AADM: dbGAP: phs001844. Requests for access to data will be granted for all research consistent with the consent provided by participants. The genome-wide association summary statistics data used in this study are publicly available at https://www.ebi.ac.uk/gwas/downloads/summary-statistics and dbGAP accession phs001672.v2.p1. The processed data generated in this study are provided in the Supplementary Information and Supplementary Data.

## Code availability

We used publicly available software GEMMA,GWAMA, Metasoft, METAL, JAM and flashfm and and PRSice-2 code is publicly available at https://github.com/genetics-statistics/GEMMA, https://bio.tools/GWAMA, https://github.com/statgen/METAL, https://www.zarlab.xyz/tag/metasoft/, https://github.com/USCbiostats/hJAM, https://github.com/jennasimit/flashfm. https://github.com/choishingwan/PRSice, Other software programs used are listed and described in the Methods.

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

## Acknowledgements

The authors would like to acknowledge all participants who contributed to the contributing cohort, including the Uganda Genome Resource, AWI-Gen, AADM, ARK, and MEIRU. The authors also acknowledge all field workers who contributed to data and sample collection. The study is funded by UKRI/MRC grant MR/Z504853/1 and Wellcome Fellowship grant 220740/Z/20/Z to Segun Fatumo. This work was supported by the UK Medical Research Council (MRC) and the UK Department for International Development (DFID) under the MRC/DFID Concordat agreement, through core funding to the MRC/UVRI and LSHTM Uganda Research Unit. A.B.K. is supported by the National Institutes of Health/ National Heart, Lung and Blood Institute (DSI-Africa grant 5U01HL172180) at the Medical Research Council/Uganda Virus Research Institute (MRC/UVRI) and LSHTM (London School of Hygiene & Tropical Medicine Uganda Research Unit). We thank the Fogarty International Center of the National Institutes of Health of the United States for supporting Sounkou Mahamane Toure under grant U2RTW010673 of the West African Center of Excellence for Global Health Bioinformatics Research Training. The authors thank Million Veteran Program (MVP) staff, researchers, and volunteers, who have contributed to MVP, and especially participants who previously served their country in the military and now generously agreed to enrol in the study. (See https://www.research.va.gov/mvp/ for more details). The citation for MVP is Gaziano, J.M. et al. Million Veteran Program: A mega-biobank to study genetic influences on health and disease. J Clin Epidemiol 70, 214-23 (2016). This research is based on data from the Million Veteran Program, Office of Research and Development, Veterans Health Administration, and was supported by the Veterans Administration (VA) Cooperative Studies Program (CSP) award #G002. "Data was accessed through approved dbGaP proposal #30287 entitled, "Genomic determinant of Complex Diseases in African ancestry individuals". We thank the Malawi Epidemiology and Intervention Research Unit, Lilongwe/Karonga, Malawi; South African Medical Research Council (with funds received from the South African National Department of Health) and the UK Medical Research Council (with funds received from the UK Government's Newton Fund) (MRC-RFA-SHIP01/2015) for the Evolving Risk Factors for Cancers in African populations study (ERICA-SA). CT is funded by the American Heart Association Early Faculty Independence (AHA/23SCEFIA1156913) and NHLBI Early Career Award (NHLBI-K01/1K01HL173644-01). JA was supported by UK Medical Research Council (MR/R021368/1, MC_UU_00002/4), Isaac Newton Trust, and Medical Research Foundation (MRF-DA-111). The AADM study is supported in part by the Intramural Research Program of the National Institutes for Research on Genomics and Global Health (CRGGH). The CRGGH is supported by the National Human Genome Research Institute, the National Institute of Diabetes and Digestive and Kidney Diseases, the Center for Information Technology, and the Office of the Director at the National Institutes of Health (NIH;1Z1AHG200362).

## Author contributions

S.F. conceptualized the study. A.B.K., J.T.B., T.C. and G.C. performed the main analyses. O.O., T.M., F.Z., R.M., S.T., O.S., C.K. and M.N. performed others post GWAS analyses. A.B.K. wrote the first draft of the manuscript. M.K., A.K., R.K., N.M., B.S., O.N., M.C., C.R.C., N.F., C.P., A.K., D.N., C.L., C.T., M.N., A.P.M., J.A., E.Z., C.R., M.R., A.A., J.F., A.C.C. and JTB, SF read and reviewed the first draft and provided critical feedback on the paper. S.F., M.R., J.F., M.C., T.C., A.A., C.R., A.P.M. supervised the work.

## Competing interests

At the time of writing, M.C. is associated with Cambridge Precision Medicine Limited, UK. However, the remaining authors declare no competing interests.

## Additional information

[1]Medical Research Council, Uganda Virus Research Institute and London School of Hygiene and Tropical Medicine (MRC/UVRI &LSHTM), Entebbe, Uganda. [2]Precision Healthcare University Research Institute, Queen Mary University of London, London, UK. [3]Malawi Epidemiology and Intervention Research Unit, Lilongwe, Malawi. [4]Department of Non-Communicable Disease Epidemiology (NCDE), London School of Hygiene and Tropical Medicine, London, UK. [5]Sydney Brenner Institute for Molecular Bioscience, Faculty of Health Sciences, University of the Witwatersrand, Johannesburg, South Africa. [6]MRC/Wits Developmental Pathways for Health Research Unit, Department of Paediatrics, Faculty of Health Sciences, University of the Witwatersrand, Johannesburg, South Africa. [7]Channing Division of Network Medicine, Department of Medicine, Brigham and Women's Hospital and Harvard Medical School, Boston, MA, USA. [8]Division of Genetics, Department of Medicine, Brigham and Women's Hospital and Harvard Medical School, Boston, MA, USA. [9]Center for Research on Genomics and Global Health, National Institutes of Health, Bethesda, MD, USA. [10]Department of Genetics, Perelman School of Medicine, University of Pennsylvania, Philadelphia, PA, USA. [11]Hatter Institute for Cardiovascular Diseases Research in Africa (HICRA), Department of Medicine, University of Cape Town, Cape Town, South Africa. [12]MRC Biostatistics Unit, University of Cambridge, Cambridge, UK. [13]Gladstone Institutes of Data Science and Biotechnology, Gladstone Institute, San Francisco, CA, USA. [14]African Center of Excellence in Bioinformatics, University of Sciences, Techniques and Technologies of Bamako, Bamako, Mali. [15]Institute of Translational Genomics, Helmholtz Zentrum München – German Research Center for Environmental Health, Neuherberg, Germany. [16]Makerere University, Kampala, Uganda. [17]Copenhagen Health Complexity Center, Department of Public Health, University of Copenhagen, Copenhagen, Denmark. [18]Nigerian Institute of Medical Research, Lagos, Nigeria. [19]Center for Genomics Research and Innovation, National Biotechnology Development Agency, Abuja, Nigeria. [20]College of Liberal Arts and Sciences, University of Westminster, London, UK. [21]Cambridge Precision Medicine Limited, ideaSpace, University of Cambridge Biomedical Innovation Hub, Cambridge, UK. [22]Division of Nephrology, Department of Medicine, Vanderbilt University Medical Center, Nashville, TN, USA. [23]The University of North Carolina at Chapel Hill, Chapel Hill, NC, USA. [24]Institute for Biomedicine, Eurac Research, Bolzano, Italy. [25]Institute of Genetic Epidemiology, Faculty of Medicine and Medical Center – University of Freiburg, Freiburg, Germany. [26]Berlin Institute of Health at Charité, Berlin, Germany. [27]Gladstone Institutes of Data Science and Biotechnology, Gladstone Institute, San Francisco, USA. [28]Department of Epidemiology and Biostatistics, University of California, San Francisco, San Francisco, CA, USA. [29]Centre for Genetics and Genomics Versus Arthritis, University of Manchester, Manchester, UK. [30]TUM School of Medicine and Health, Technical University of Munich (TUM), TUM University Hospital, Munich, Germany. [31]Medical Research Council/Wits University Rural Public Health and Health Transitions Research Unit (Agincourt), School of Public Health, Faculty of Health Sciences, University of the Witwatersrand, Johannesburg, South Africa. [32]Wits Donald Gordon Medical Research Institute, Faculty of Health Sciences, University of the Witwatersrand, Johannesburg, South Africa. [33]School of Global and Public Health, Kamuzu University of Health Sciences, Blantyre, Malawi. [34]School of Health and Wellbeing, University of Glasgow, Glasgow, UK. [35]Epidemiology and Population Health, London School of Hygiene and Tropical Medicine, London, UK. [36]Strengthening Oncology Services Research Unit, Faculty of Health Sciences, University of the Witwatersrand, Johannesburg, South Africa. [37]These authors contributed equally: Abram B. Kamiza, Tinashe Chikowore. ✉e-mail: s.fatumo@qmul.ac.uk

