## [Transparent Peer Review file · Nature Communications]

KidneyGenAfrica multi-cohort genome-wide association identifies novel genetic loci for kidney function in Africa

Corresponding Author: Professor Segun Fatumo

Version 0:

Reviewer comments:

Reviewer #1

(Remarks to the Author)

Kamiza and Chikowore et. al. performed meta-GWAS for eGFR (creatinine-estimated glomerular filtration rate) in 26k Africans and 81k African ancestry individuals. They compare meta-analysis results from East Africa, West Africa, South Africa, continental Africa, and all 107k African diaspora individuals. They go on to perform fine mapping, eQTL colocalization, functional enrichment, and polygenic score analyses. Strengths of the paper include one of the largest African GWAS to date, discovery of novel associations not found in European and Asian GWAS, and evidence that well known APOL1 variants do not confer kidney disease risk the same in all African ancestries. This paper will be of interest to complex trait geneticists and those interested in kidney disease. Clarifications and additions that could improve the paper include:

1. Do the distributions of eGFR vary among populations/regions? How may this affect your results?
2. More details about how related individuals were excluded or included are needed (line 146...). What IBD thresholds were used? Did your principal component analysis account for relatedness with PC-AiR or similar?
3. The African Genome Resources used by the Sanger Imputation Service are comprised primarily of Ugandan individuals. How might this affect your downstream analyses in other populations?
4. Similarly, were the APOL1 variants used to define haplotypes genotyped directly on the array or imputed? If imputed, can you confirm similar imputed genotypes with other reference panels like 1000G or TopMed? Could poorer imputation explain the South and West African results in Table 4?
5. In Fig 3d, I suggest calculating PCs with only African reference populations to more easily view genetic distances. If related individuals are included, I suggest using PC-AiR (<https://rdrr.io/bioc/GENESIS/man/pcair.html>). It is challenging to distinguish so many colors. Also, please make the font larger in all your Fig 3 plots and explain what the abbreviations stand for in the legend.
6. The methods lines 277-280 state “The generated PGSs were incorporated into the linear regression model to explain eGFR and performance, while adjusting for age, sex, and the five principal components as fully adjusted model. For the null model, we only included PGSs. R2 is the difference between the fully adjusted and null model.” In most PGS work, the null model includes all covariates with no PGS and is compared to the fully adjusted (PGS + all covariates) model. Are the methods incorrectly stated, or should you redo the PGS analyses presented in Fig 3c?
7. Cross-population polygenic score methods like PRS-CSx and CT-SLEB have been shown to perform better than the clumping and thresholding method of PRSice-2 (ref 74 and <https://www.nature.com/articles/s41588-022-01054-7>) A more fair comparison for multi-ancestry PGS compared would use PRS-CSx or CT-SLEB rather than C+T.
8. Line 399 of the discussion indicates that your colocalization analyses assumed a single causal variant. Susie-coloc allows for multiple causal variants (https://chr1swallace.github.io/coloc/articles/a06_SuSiE.html) and could identify additional colocalized variants.

Minor

1. Line 378 is redundant with line 373.
2. Define the FSGS abbreviation at line 384.

Reviewer #2

(Remarks to the Author)

In their work, Kamiza, Chikowore and colleagues describe a novel continental consortium effort to study the genetics of kidney function among African-ancestry individuals through GWAS. In the so-named KidneyGenAfrica consortium, the authors collected GWAS data for estimated glomerular filtration rate (eGFR) from Eastern, Western and Southern African cohorts, followed by regional GWAS meta-analyses and continental GWAS meta-analyses, and finally global GWAS meta-analyses with existing datasets (notably, Million Veterans Program and CKDGen). The regional analyses identified several novel loci, although these did not replicate across regions nor within the continental analysis (which seemingly only identified one known locus). The global meta-analysis identified 20 independent loci, of which 3 were novel for kidney function. The authors went on to perform various, quite standard, post-GWAS analyses such as molecular trait colocalization, pathway enrichment analyses, and PheWAS. The authors also specifically assessed the well-described APOL1 risk haplotypes, and concluded that the haplotype conferred far weaker effects on kidney function compared to previous reports. Finally, the authors built polygenic scores (PGS) from their GWAS data, and showed that African-specific PGS showed the best prediction in held-out African datasets, even when compared to existing multi-ancestry PGS.

The consortium and findings are potentially of very high impact to the field: The work represent one of the first large GWAS studies that included individuals from a wide variety of regions across Africa (and the world). Considering the current dominance of European genetic ancestry in GWAS world, such efforts have the potential to transform genomics, not only for novel discovery but also for improved and more equitable genetic risk prediction.

However, as it stands, I remain unsure of the quality of the data and the associated conclusions. I therefore have several major comments/questions that will need to be answered before the work is publishable, in my mind.

1. The first major issue is the apparent non-replication of most novel regional loci. As I read the results, these loci do not appear in any other larger subsequent meta-analysis, and don't replicate across regional data. The authors interpret this as evidence for strong regional heterogeneity. I think this is a very optimistic claim, and one that does not align with previous extensive (albeit European) GWAS data. More likely it would seem that these loci represent false-positives, which obviously would pull into doubt all subsequent meta-analyses. Can the authors provide any data to support those loci as real?
2. Potentially related to the above, I worry about the imputation reference panel used by the authors for their study. It is absolutely fantastic to see such a massive African-centric consortium, but why then use a small (~2k) African-American dataset as imputation reference? I am concerned that the imputation reference does not match the samples well enough, which might in turn explain the unexpected APOL1 results. Are there any Continental-African WGS-based datasets the authors can use to assess the impact of their imputation reference? At least to assess the accuracy of imputation for the APOL1 haplotypes?
3. I also think the discussion can be substantially improved. It seems to me that far too much weight is put on the PheWAS results; in fact, the authors make several claims that seem to border on causality, even though PheWAS only really tells you about pleiotropy. The PheWAS-based discussion needs to be substantially shortened and softened. At the same, discussion about the truly novel loci – which were not identified in previous European / multi-ancestry studies – should be more prominently discussed.

I also have some more minor comments to consider:

METHODS

4. Why use random-effects meta-analyses for within-Africa regional meta-analyses, but use sample size-weighted fixed-effects meta-analyses for the pan-African-ancestry meta-analyses? On that note, perform random-effects meta-analysis across regions, but subsequently fixed-effects meta-analyses in the global meta-analysis? How would each of the meta-analyses have compared if the authors used simple inverse-variance weighted fixed-effects meta-analyses across the board?
5. The coloc analyses with molecular traits are interesting. However, please comment on the ancestry of the molecular trait data (GTEx, NephthQTL) and discuss how ancestry mis-match might affect the results. Please specify how the LD reference data were handled in this context.
6. In the PGS analyses, I am a bit confused about what meta-analyses are used for what testing set (ie, when were the pan-African-ancestry meta-analyses used, and when only the continental meta-analyses?). Please make this more clear.
7. "R2 is the difference between the fully adjusted and null model" should this not be reported as the the difference between the fully adjusted and null model divided by the residual variance, so $(R2_{full} - R2_{null}) / (1 - R2_{null})$?

RESULTS

8. Please also briefly summarize what loci are not novel (to better highlight whether the stronger known loci are recapitulated in your African GWAS analysis).
9. Please more clearly highlight what the inflation factors and LDSC intercepts were for each of the regional analyses.
10. As mentioned in the major comments, the lack of replication across the African regions, as described by the authors, is rather worrisome. However, it is not clear whether the authors simply mean that the loci are not genome-wide significant in the other regions, or whether the loci show no association at all. This should be more clearly defined. I.e, what does the overlap look like when you take a liberal cutoff like $P < 1e-4$ for these loci?
11. The authors report relatively lower effect sizes of APOL1 variants on kidney function as compared to previous publications. Assuming the APOL1 variants are well-imputed, I think the authors should better discuss how phenotyping might affect the associations. Notably, it should be noted that the authors defined low kidney function as being under a certain eGFR threshold, whereas previous studies typically used EHR diagnoses as outcomes. Please discuss how this might impact the effect size estimation.
12. In the results, it might be worth highlighting more prominently that the much larger multi-ancestry PGS performed worse than African-specific PGS. Also please discuss some of the R^2 values in the text, so the reader can better follow the story.

DISCUSSION

13. "In our analysis, we noted that rs6670659, rs74383679, and rs7763270 were common genetic variants in Africa" it is probably helpful to also highlight the nearby gene so the reader better understands the associated loci.
14. Please also discuss how the various ascertainment procedures across the various cohorts might affect the GWAS results.

Reviewer #3

(Remarks to the Author)

This study presents a GWAS meta-analysis on estimated glomerular filtration rate (eGFR) across multiple African populations. The authors conduct regional, continental, and pan-African meta-analyses. While this study aims to address the underrepresentation of African populations in GWAS, I have major concerns regarding the robustness of the findings.

Major Concerns:

1. Lack of replication for novel variants

One of the main concerns is that the novel variants identified in the regional meta-analyses fail to replicate in other GWAS included in the study. In standard GWAS practice, replication in independent datasets is essential to validate findings and minimize false positives. The authors mention a "lack of overlapping genetic loci," but they do not provide specific association results, significance thresholds, or statistical replication power. Without these details, it is unclear whether the variants are nominally significant ($P < 0.05$) but fail to reach genome-wide significance, or if they show no signal at all.

The fact that these novel regional variants do not emerge in the pan-African meta-analysis is particularly concerning. If these variants were truly region-specific, their absence of association elsewhere would be expected. However, upon examining the UK Biobank African ancestry subgroup, I found that all these variants exist at similar allele frequencies as reported in Table 1. This suggests that their absence in replication efforts is unlikely to be due to regional specificity. If these variants genuinely influence eGFR, they should be detectable in larger independent datasets with sufficient statistical power. The authors need to investigate and explain why these associations fail to replicate.

To improve transparency, the authors should provide association results for all reported variants across each meta-analysis. Additionally, they should clarify whether their statements regarding "lack of overlap" refers to the lack of presence of these variants in different populations (which is doubtful, given their presence in the UK Biobank) or to differences in association results. Throughout the manuscript the authors do not say what significance thresholds they are using when discussing replication. For example, in line 379, when the authors state that "none was found to be significant in the regional and continental meta-analyses," it is unclear what significance threshold is being used.

2. Lack of individual GWAS results in meta-analyses

The manuscript presents findings from meta-analyses but does not report results from the individual GWAS that contribute to each meta-analysis. This omission makes it impossible to determine whether the reported signals are robust across all datasets or driven by a single study. A well-conducted meta-analysis should demonstrate consistency of effect sizes across studies; otherwise, associations driven by only one dataset are more likely to be false positives.

To address this, the authors should provide forest plots or heterogeneity statistics to illustrate the consistency of effects across contributing GWAS. These analyses would clarify whether the reported associations are truly replicated across datasets or are disproportionately influenced by a single study. Without this information, it is difficult to assess the reliability of the reported associations.

3. Claimed novel loci close to GATM

The manuscript identifies rs12595073 (SORD2P) and rs1918516 (SQRDL) as novel loci, but both variants are in close proximity to GATM locus, which is one of the strongest eGFR-associated signal in the genome. Given their location, these variants most likely reflect linkage disequilibrium with the well-established GATM association rather than representing independent signals. The authors present regional locus plots with a 500kb window, but a broader view would reveal that these loci are at the “tails” of the GATM signal. After conditioning on all independent GATM associated variants, it is doubtful that these variants would remain significant. The authors identify independent loci by “clumping 500kb around lead SNPs” but in this instance the window needs to be extended further.

4. No comparison with European allele frequencies or GWAS results

The manuscript would benefit from comparison between the reported variants and European GWAS findings. Specifically, reporting European minor allele frequencies (MAFs) for the variants identified in the pan-African meta-analysis and providing effect size comparisons or scatter plots would help contextualize these findings. Understanding how these associations compare across ancestries could provide insights into potential genetic and environmental influences on eGFR. It is not clear which of the identified variants are African specific.

Other points:

The abstract claims 28 loci and 5 novel variants from the regional meta-analysis. In the results text and Table 1 there are only 7 loci. Where are the other 21 loci?

The abstract and results text claims 20 loci from the pan-African meta analysis, but Table 2 only has 19 variants.

The authors claim 3 novel variants and point to Fig.S2 which has locus plots for four variants. I do not know why the fourth locus plot for 11:5513654 is there.

Figures have small text that is very difficult to read.

The statement that the distribution of APOL1 variants 'demonstrates the diversity' of African populations might downplay the well-established genetic diversity of African populations. Reframing this to focus on the specific findings about APOL1 variants, within the broader context of known diversity, could improve clarity and avoid any potential for misunderstanding.

Version 1:

Reviewer comments:

Reviewer #1

(Remarks to the Author)

The authors have satisfactorily addressed my major concerns. Thank you.

According to the summary form: “The genome-wide association study summary statistic data generated in this study have been deposited in the GWAS Catalog database under accession code [add hyperlink here].” Is the link available? Please include a Data Availability statement and accessions in the final manuscript.

Minor comments:

1. Make it clear in the Fig. 3c legend that the numbers on top of the bars are p-values.
2. Please confirm the best R² PRS performance was R²=0.11%=0.0011 rather than R²=0.11. The manuscript conflicts with the response to reviewer 2, item 7.

Reviewer #2

(Remarks to the Author)

The authors have responded to many of my comments appropriately.

Nevertheless, a major issue regarding the consistency of the novel regional-specific loci remains.

Notably, the authors argue that replication of these loci was not possible due to low allele frequencies in existing European and Asian ancestry datasets, which is a fair point. However, my comment did not pertain to replication within other major ancestries, but pertained to the consistency within the various regional African meta-analysis done by the authors. The regional-specific loci do not seem to replicate meaningfully _in the other regional African GWAS_. This is an important point, as it could potentially mean that the regional-specific loci are not robust, and I think the authors need to be more transparent here. What were the allele frequencies of these loci in all regional meta-analyses? And what were the association results? If well-enough powered, one would expect some replication signal across African regions, unless the signals represent artifacts.

I should note that the authors do present a forest plot with – seemingly – the regional association results, although this figure

does not seem to be referenced in the associated text and does not have a legend, making it difficult to understand. That being said, the weird patterns of association - for several of the loci – makes me concerned that harmonization of effect alleles was improperly performed in each sub-stratum prior to meta-analysis. For instance, it is very strange that 3 strata show significant positive effects for rs1706775, while two strata show nominally significant results in the opposite direction. The same seems to apply to rs6670659. On the one hand, this could explain the loss of significance in the larger meta-analysis (if the loci are actually bona fide), and on the other hand it should motivate a careful re-assessment of the harmonization steps taking prior to the meta-analyses.

Reviewer #3

(Remarks to the Author)

Major Concerns

The newly provided tables and forest plots reinforce my concern that the seven regional lead SNPs are likely false positive associations rather than true, region-specific signals.

The forest plots for rs1706775 and rs6670659 reveal opposing effect directions across cohorts, yet the meta-analysis still produces highly significant associations, which does not make sense. For the other 5 variants, one dataset is mostly driving the association.

The authors refer to “replication within contributing studies,” but this is not replication in the GWAS sense. True replication requires observation of a consistent signal in an independent dataset or population. The critical issue is that within Africa, these signals do not replicate across regions or in independent African ancestry datasets (e.g., MVP or UK Biobank), despite similar allele frequencies.

Without evidence of replication, either across regions or in other cohorts, and given the visible inconsistencies in the forest plots, there is no statistical or biological basis to present these loci as robust associations. Furthermore, no heterogeneity statistics are provided to assess consistency (e.g., I^2 , Q).

At present, none of these seven loci meet the standard replication or robustness criteria expected of GWAS discoveries.

Additionally, two of the claimed novel loci (rs12595073 and rs1918516) are located immediately upstream and downstream of the well-established GATM locus. Given their proximity and the strength of the GATM signal, it is highly likely they reflect the extended association at this locus rather than independent signals. The appropriate analysis is to test whether these variants remain associated with eGFR after adjusting for the GATM variants. The author provided the conditional association the other way around. Without correct conditional analysis, these loci should not be presented as novel.

Version 2:

Reviewer comments:

Reviewer #1

(Remarks to the Author)

The authors have addressed my concerns. I believe the edits they have made highlight the challenges of replicating associations across the diverse genetic and environmental backgrounds of regional groups across Africa.

Reviewer #2

(Remarks to the Author)

No further comments.

REVIEWER COMMENTS

Reviewer #1 (Remarks to the Author):

Kamiza and Chikowore et. al. performed meta-GWAS for eGFR (creatinine-estimated glomerular filtration rate) in 26k Africans and 81k African ancestry individuals. They compare meta-analysis results from East Africa, West Africa, South Africa, continental Africa, and all 107k African diaspora individuals. They go on to perform fine mapping, eQTL colocalization, functional enrichment, and polygenic score analyses. Strengths of the paper include one of the largest African GWAS to date, discovery of novel associations not found in European and Asian GWAS, and evidence that well known APOL1 variants do not confer kidney disease risk the same in all African ancestries. This paper will be of interest to complex trait geneticists and those interested in kidney disease. Clarifications and additions that could improve the paper include:

1. Do the distributions of eGFR vary among populations/regions? How may this affect your results?

Response: Thank you for your thoughtful comments. Yes, the distribution of eGFR varied by region (West, East, South) as shown in the boxplot below. Nevertheless the 25th -75th percentile interquartile range distribution of the eGFR were overlapping among the east, west and south. To further minimize the potential impact of these differences we applied the same transformation on the eGFR, and the distribution were the same in each contributing studies and cohort see figure below. We then stratified the analysis by region by adopting a three stage meta-analysis. We then performed meta-analysis using software that account for heterogeneity. In the first stage we performed a regional meta-analysis using a fixed effect method implemented in GWAMA¹. We used this method, with the assumption that the distribution of eGFR within the geographical region is not different. In the second stage, we performed continental meta-analysis, using Han and Eskin random effect methods implemented in Metasoft to account for the heterogeneity ². In the third stage, we used the sample weighted methods for the pan-African meta-analysis using the approach implemented in META³. We used this approach because studies like CKDGen, MVP and UKBB have used different transformations, and the effect sizes are therefore not on the same scale. In this revision, we have added the following statement on lines 160-162 “eGFR was calculated using the CKD-EPI equation without using the coefficient for race-based adjustment. The distribution of eGFR varied by the geographical region (**Fig.S1a**), we then inverse-rank normal transformation of eGFR (**Fig.S1b**).”

Reference

1. Mägi, R. & Morris, A. P. GWAMA: software for genome-wide association meta-analysis. *BMC Bioinformatics* **11**, 288 (2010).
2. Han, B. & Eskin, E. Random-effects model aimed at discovering associations in meta-analysis of genome-wide association studies. *Am J Hum Genet* **88**, 586–598 (2011).
3. Willer, C. J., Li, Y. & Abecasis, G. R. METAL: fast and efficient meta-analysis of genomewide association scans. *Bioinformatics* **26**, 2190–2191 (2010).

(a) CKD-EPI equation without using the coefficient for race-based adjustment (b) the inverse ranked eGFR transformation

2. More details about how related individuals were excluded or included are needed (line 146...). What IBD thresholds were used? Did your principal component analysis account for relatedness with PC-AiR or similar?

Response: Thank you for your comment. Potential high related individuals were removed with identity by descent (IBD) PIHAT > 0.9 in all participating cohorts and studies. Moreover, to account for potential remaining cryptic relatedness we used GEMMA as it implements a mixed model and is known to be more efficient in correcting for high relatedness. We have included these statements on this in lines 193-196 and 212-216 in the Methods section.

On lines 193-196 “Sample and SNP quality controls (QC) were performed for all genotyped data from the participating cohorts and studies in continental Africa. We excluded individuals who were highly related with the identity by descent (IBD) PIHAT > 0.9, had discordant sex information between the genotyped and phenotype data, had high genotype missingness, and an outlying heterozygosity rate”

On lines 212-216 “For association testing, we used a linear mixed model implemented in GEMMA²⁴, using the dosage format of the genotype data. The genomic relationship matrices (GRMs) were calculated on the pruned dataset from each contributing cohort (filtered for MAF < 0.01,

missingness < 0.05, and pruned in Plink with the option --indep-pairwise 50 5 0.2). The GRM was included in the linear mixed model to adjust for the remaining relatedness among individuals of African ancestry. “

3. The African Genome Resources used by the Sanger Imputation Service are comprised primarily of Ugandan individuals. How might this affect your downstream analyses in other populations

Response: Thank you for your comment. The African Genome Resources (AGR) includes individuals from all three major geographical regions of Africa and represents diverse ethnolinguistic groups, including the Khoe-San, Nilo-Saharan, Niger-Congo, and Afro-Asiatic populations¹. The inclusion of these individuals in the African genome Resource increased the number of SNPs and overall imputation coverage.

Reference

1 Sengupta, D. et al. Performance and accuracy evaluation of reference panels for genotype imputation in sub-Saharan African populations. *Cell Genom* **3**, 100332 (2023).

4. Similarly, were the APOL1 variants used to define haplotypes genotyped directly on the array or imputed? If imputed, can you confirm similar imputed genotypes with other reference panels like 1000G or TopMed? Could poorer imputation explain the South and West African results in Table 4?

Response: We have carefully assessed the imputation quality of the APOL1 variants used in our study and found no evidence of poor imputation. The South African group in Table 4 includes participants from both the MEIRU and AWI-Gen South African cohorts, while the West African group comprises the AWI-Gen West Africa cohort (Burkina Faso and Ghana). Comparisons between the MEIRU cohort and the AWI-Gen South African sites showed similar APOL1 allele distributions.

Importantly, we evaluated imputation performance by comparing imputed APOL1 variants with whole-genome sequencing (WGS) data from the AWI-Gen cohort. rs73885319 (G1) was directly genotyped on the H3Africa SNP array and demonstrated 100% concordance between genotyped and imputed calls. The other two key variants, rs60910145 (p.I384M) and rs71785313 (p.N388Y389del), were imputed and showed 100% and 99% concordance, respectively, when compared with AWI-Gen WGS data¹

We compared these variants across multiple reference datasets to further validate imputation accuracy, including the 1000 Genomes African samples, the African Genome Variation Project (AGVP), and the 1000 African Genomes project. Concordance for rs73885319 was 100% across all datasets; rs60910145 showed 99.9%–100% concordance, and rs71785313 showed up to 99% concordance. For consistency across cohorts in our APOL1 haplotype analysis, we used a proxy SNP, rs12106505, to define the G2 allele, which showed 98% concordance with rs71785313 in AWIGEN South Africa.

Given these high concordance rates, we are confident that imputation quality is robust and does not explain the lack of association between APOL1 variants and low eGFR in Southern and West African cohorts. However, it is important to note that APOL1 allele frequencies are known to vary by population and geography. The West African cohort from Ghana (Navrongo) and Burkina Faso (Nanoro) has a lower frequency of G1 and G2 than other West populations. These populations

also have fewer cases of low eGFR, so taken together, the West African populations reported will have less power to detect an association. In other words, our findings reflect the populations sampled and may not represent the entire West African region. Previous studies have reported substantial heterogeneity in APOL1 allele ^{2,3}. For example, G2 allele frequency within West Africa—for example, 6–16.7% among the Yoruba in Nigeria, and 6–24.4% across West African populations ⁴, highlighting the influence of demographic structure on APOL1 distribution. Detection of APOL1 with a stronger effect size is mostly seen in end-stage kidney disease, focal segmental glomerulosclerosis, and HIV-associated nephropathy studies ⁵

We have added this statement to the result and discussion sections:

Result section, lines 370-373

Our Southern African and West African cohorts (from Ghana – Navrongo and Burkina Faso - Nanoro) have fewer cases of individuals with low eGFR and APOL1 high-risk alleles which could influence the statistical power to detect an association. This could further show the effect size difference of APOL1 risk alleles in a population-based study compared to a case-control study.

In the discussion section, lines 440-456

Substantial heterogeneity in APOL1 alleles (for example, G2 allele frequency within West Africa, for example, 6–16.7% among the Yorubas in Nigeria, and 6–24.4% across West African populations) has been reported in previous studies, highlighting the influence of demographic structure on APOL1 distribution.

In the study of Gbadegesin et al., the authors reported an association between APOL1 variants and chronic kidney disease (CKD) in cohorts from Nigeria and Ghana. Their definition of CKD included eGFR <90 mL/min/1.73 m², a urinary albumin-to-creatinine ratio ≥30 mg/g, or both. In addition, a proportion of their cases included individuals with biopsy-confirmed glomerular disease and those with sickle cell disease, a known risk factor for kidney impairment. In contrast, our study used a cross-sectional, population-based approach relying on a single measurement of eGFR to define low eGFR. While eGFR is a useful indicator of renal function at the population level, it is not the same as a clinical diagnosis of CKD, especially in the absence of longitudinal data. Thus, our classification of kidney function status for APOL1 analysis, the demography of participants, and study design may contribute to the lower effect sizes observed in our study compared to previous findings. Detection of APOL1 with a stronger effect size is mostly seen in end-stage kidney disease, focal segmental glomerulosclerosis, and HIV-associated nephropathy studies.

References:

1 Brandenburg JT, Govender MA, Winkler CA, et al. Apolipoprotein L1 High-Risk Genotypes and Albuminuria in Sub-Saharan African Populations. *Clin J Am Soc Nephrol.* 2022;17(6):798-808. doi:10.2215/CJN.14321121

2 Ko WY, Rajan P, Gomez F, et al. Identifying Darwinian selection acting on different human APOL1 variants among diverse African populations [published correction appears in *Am J Hum Genet.* 2013 Jul 11;93(1):191. An, Ping [added]; Winkler, Cheryl A [added]]. *Am J Hum Genet.* 2013;93(1):54-66. doi:10.1016/j.ajhg.2013.05.014

3 Tzur, S., Rosset, S., Shemer, R. et al. Missense mutations in the APOL1 gene are highly associated with end stage kidney disease risk previously attributed to the MYH9 gene. *Hum Genet* **128**, 345–350 (2010). <https://doi.org/10.1007/s00439-010-0861-0>

4 Ulasi II, Tzur S, Wasser WG, et al. High population frequencies of APOL1 risk variants are associated with increased prevalence of non-diabetic chronic kidney disease in the Igbo people from south-eastern Nigeria. *Nephron Clin Pract.* 2013;123(1-2):123-128. doi:10.1159/000353223

5 Yusuf AA, Govender MA, Brandenburg JT, Winkler CA. Kidney disease and APOL1. *Hum Mol Genet.* 2021;30(R1):R129-R137. doi:10.1093/hmg/ddab024

5. In Fig 3d, I suggest calculating PCs with only African reference populations to more easily view genetic distances. If related individuals are included, I suggest using PC-AiR (<https://rdrr.io/bioc/GENESIS/man/pcair.html>). It is challenging to distinguish so many colors. Also, please make the font larger in all your Fig 3 plots and explain what the abbreviations stand for in the legend.

Response: Thank you for your comment and suggestions. In the PCs calculation, we removed related individuals with IBD PIHAT > 0.185 as well as pruned high correlated SNPs. In this revision, we have recalculated the PCs using data from individuals of African ancestry only. We have also increased the font size of the figure legend. Lastly, we have provided the meaning of the abbreviations in the figure footnote as shown below.

6. The methods lines 277-280 state “The generated PGSs were incorporated into the linear regression model to explain eGFR and performance, while adjusting for age, sex, and the five principal components as fully adjusted model. For the null model, we only included PGSs. R² is the difference between the fully adjusted and null model.”

In most PGS work, the null model includes all covariates with no PGS and is compared to the fully adjusted (PGS + all covariates) model. Are the methods incorrectly stated, or should you redo the PGS analyses presented in Fig 3c?

Response: Thank you for your comment and suggestions. The above statement is not correctly stated. In this revision, we have corrected the mistake and now the statement reads in lines 280-282 “The generated PGSs were incorporated into the linear regression model to explain eGFR. In the null model we include age, sex, and the five principal components. For the full model, we included PGSs, sex, age, and five principal components.”

7. Cross-population polygenic score methods like PRS-CSx and CT-SLEB have been shown to perform better than the clumping and thresholding method of PRSice-2 (ref 74 and <https://www.nature.com/articles/s41588-022-01054-7>) A more fair comparison for multi-ancestry PGS compared would use PRS-CSx or CT-SLEB rather than C+T.

Response: Thank you for your comment and suggestion. We fully agree that PRS-CSx and CT-SLEB have been shown to outperform the clumping and thresholding PRS method. In fact, our analysis plan intended to use PRS-CSx. However, the performance of the polygenic scores (PGS) generated with this method was suboptimal due to the limited number of overlapping SNPs between our target data and the reference data required by PRS-CSx. Our GWAS has over 13 million SNPs, whereas the HapMap3 reference data used in PRS-CSx contains only about one million SNPs, resulting in the exclusion of approximately 12 million SNPs during PGS calculation. However, with PRSice, our base and target datasets perfectly aligned, as it does not require reference data for score computation.

8. Line 399 of the discussion indicates that your colocalization analyses assumed a single causal variant. Susie-coloc allows for multiple causal variants (https://chr1swallace.github.io/coloc/articles/a06_SuSiE.html) and could identify additional colocalized variants.

Response: Thank you for comment. As noted, the single causal allele assumption may be violated in some instances, it was the practical choice in this work given the heterogeneity of LD due to within and between ancestry differences - the target GWAS was mainly composed of individuals of recent diverse African Ancestries and the main eQTL dataset, namely GTExV8, which is mainly of recent European ancestry. Given this limitation, we highlighted loci where there was evidence of violation of the assumption. Given the setup, it may be fruitful in subsequent work to investigate the application of multi-trait multi-group fine mapping ¹ methods for scenarios similar to the current one and or leverage more cosmopolitan eQTL GWAS ² with suitable LD reference and methods.

Reference

1 Zhou, F. *et al.* Leveraging information between multiple population groups and traits improves fine-mapping resolution. *Nat Commun* **14**, 7279 (2023).

2 Orchard, P. *et al.* Cross-cohort analysis of expression and splicing quantitative trait loci in TOPMed. 2025.02.19.25322561 Preprint at <https://doi.org/10.1101/2025.02.19.25322561> (2025).

Minor

1. Line 378 is redundant with line 373.

Response: Thank you for your comment. In this revision, we have corrected the redundant line at number 373.

2. Define the FSGS abbreviation at line 384.

Response: Thank you for your comment. In this revision, we have defined FSGS as focal segmental glomerulosclerosis now on line 396

Reviewer #2 (Remarks to the Author):

In their work, Kamiza, Chikowore and colleagues describe a novel continental consortium effort to study the genetics of kidney function among African-ancestry individuals through GWAS. In the so-named KidneyGenAfrica consortium, the authors collected GWAS data for estimated glomerular filtration rate (eGFR) from Eastern, Western and Southern African cohorts, followed by regional GWAS meta-analyses and continental GWAS meta-analyses, and finally global GWAS meta-analyses with existing datasets (notably, Million Veterans Program and CKDGen). The regional analyses identified several novel loci, although these did not replicate across regions nor within the continental analysis (which seemingly only identified one known locus). The global meta-analysis identified 20 independent loci, of which 3 were novel for kidney function. The authors went on to perform various, quite standard, post-GWAS analyses such as molecular trait colocalization, pathway enrichment analyses, and PheWAS. The authors also specifically assessed the well-described APOL1 risk haplotypes, and concluded that the haplotype conferred far weaker effects on kidney function compared to previous reports. Finally, the authors built polygenic scores (PGS) from their GWAS data, and showed that African-specific PGS showed the best prediction in held-out African datasets, even when compared to existing multi-ancestry PGS.

The consortium and findings are potentially of very high impact to the field: The work represent one of the first large GWAS studies that included individuals from a wide variety of regions across Africa (and the world). Considering the current dominance of European genetic ancestry in GWAS world, such efforts have the potential to transform genomics, not only for novel discovery but also for improved and more equitable genetic risk prediction.

However, as it stands, I remain unsure of the quality of the data and the associated conclusions. I therefore have several major comments/questions that will need to be answered before the work is publishable, in my mind.

1. The first major issue is the apparent non-replication of most novel regional loci. As I read the results, these loci do not appear in any other larger subsequent meta-analysis, and don't replicate across regional data. The authors interpret this as evidence for strong regional heterogeneity. I think this is a very optimistic claim, and one that does not align with previous extensive (albeit European) GWAS data. More likely it would seem that these loci represent false-positives, which obviously would pull into doubt all subsequent meta-analyses. Can the authors provide any data to support those loci as real?

Response: Thank you for your comment. We checked the minor allele frequency of the regional variants in individuals of European and Asian ancestries. We found these variants to be monomorphic in European and Asian ancestry population (see table below). Hence, they cannot be replicated in these ancestry populations. Interestingly, our variants are not artefacts as they are noted to be significant (p -value < 0.05) with the same direction of effect in more than one of contributing studies to the regional meta-analyses as shown in Figure below. Moreover, there is much more diversity across Africa than across Europe, so the lack of heterogeneity within European populations does not transfer to Africa. In this revision, we have included these changes on lines 305-307 and 394-410

Table 1. Minor allele frequency distribution of independent loci associated with eGFR in individuals of African ancestry in the regional meta-analysis													
SNP	Genes	CHR	BP	EA	NEA	EAF	BETA	SE	P-value	Region	EAF- AFR	EAF- EUR	EAF- EAS
rs6670659	HSD3BP3	1	120090766	C	G	0.769	0.11	0.019	1.24E-08	Eastern	0.27	0	0
rs74383679*	FAM72C	1	206198189	G	C	0.155	0.159	0.028	2.75E-08	Southern	0.16	0	0
rs7763270*	LAMA4	6	112537967	A	G	0.139	-0.193	0.034	2.95E-08	Western	0.13	0	0
rs73788952*	OPRM1	6	154342741	G	A	0.092	0.132	0.022	5.04E-09	Southern	0.1	0	0
rs115943222*	KLHL1	13	70304786	A	C	0.009	-0.505	0.092	4.72E-08	Southern	0.03	0	0
rs1706775	SLC28A2	15	45586652	T	C	0.572	0.133	0.015	3.85E-17	Eastern	0.4	0	0
rs4243062*	LOC645752	15	78189084	T	C	0.299	-0.101	0.018	3.19E-08	Eastern	0.27	0.06	0.13

SNP; single nucleotide polymorphism, CHR chromosome, BP, base pair position, EA; effect allele, NEA, non-effect allele, EAF; Effect allele frequency, SE; standard error, * novel loci, AFR; African, EUR; European, EAS; East Asian ancestry

2. Potentially related to the above, I worry about the imputation reference panel used by the authors for their study. It is absolutely fantastic to see such a massive African-centric consortium, but why then use a small (~2k) African-American dataset as imputation reference? I am concerned that the imputation reference does not match the samples well enough, which might in turn explain the unexpected APOL1 results. Are there any Continental-African WGS-based datasets the authors can use to assess the impact of their imputation reference? At least to assess the accuracy of imputation for the APOL1 haplotypes?

Response: Thank you for your comment and suggestion. All participating cohorts and studies in the KidneyGen Africa project were imputed using the African Genome Resource (AGR) reference panel available on the Sanger Imputation Server. Currently, the AGR is the largest and only imputation reference panel specifically designed for continental Africa. This panel is primarily composed of individuals from all three main geographical regions in Africa including the Khoesan, Nilo-Saharan, Niger-Congo and Afro-Asiatic¹, and it is not an imputation reference panel of African Americans.

Reference

1 Sengupta, D. et al. Performance and accuracy evaluation of reference panels for genotype imputation in sub-Saharan African populations. *Cell Genom* **3**, 100332 (2023).

3. I also think the discussion can be substantially improved. It seems to me that far too much weight is put on the PheWAS results; in fact, the authors make several claims that seem to border on causality, even though PheWAS only really tells you about pleiotropy. The PheWAS-based discussion needs to be substantially shortened and softened. At the same, discussion about the truly novel loci – which were not identified in previous European / multi-ancestry studies – should be more prominently discussed.

Response: Thank you for your comment and suggestions. In this revision, we have shorted the discussion on the PheWAs see lines 425-430. “Our PheWAS revealed biologically relevant associations beyond eGFR, highlighting the impact of lead variants from our pan-African meta-analyses on various metabolic traits, including CKD and related risk factors. We observed significant links to common comorbidities such as cardiovascular conditions (e.g., obesity, hypertension, diabetes) and immunological traits (e.g., red blood cell characteristics). Notably, we found associations with dermatological traits (ease of skin tanning and skin colour) and psychiatric traits (depression, sleep patterns, alcohol, and tobacco use)^{51–57}.” We also took this opportunity to discuss the results of our novel and independent loci in detail see line 392-410 of this revision.

I also have some more minor comments to consider:

METHODS

4. Why use random-effects meta-analyses for within-Africa regional meta-analyses, but use sample size-weighted fixed-effects meta-analyses for the pan-African-ancestry meta-analyses? On that note, perform random-effects meta-analysis across regions, but subsequently fixed-effects meta-analyses in the global meta-analysis? How would each of the meta-analyses have compared if the authors used simple inverse-variance weighted fixed-effects meta-analyses across the board?

Response: Thank you for your insightful comments and suggestions. Our assumption was low heterogeneity within a geographical region in Africa, and we performed a fixed-effect meta-analysis using the GWAMA software. For the continental African meta-analyses, we applied the Han and Eskin random-effects model implemented in METASOFT, considering the high heterogeneity among the three geographical regions in Africa. For the pan-African meta-analysis, we used Stouffer's method, which only uses p-value and direction of effect to perform a meta-analysis and does not make use of effect sizes. We chose this approach because different transformations were used by studies in the pan-Africa meta-analysis, meaning that effect sizes were on different scales. Moreover, we have listed the different transformation of eGFR as one of the limitations of the study on lines 482-485 reads "However, our study was limited by ancestry mismatch between our data and the GTEx data used for colocalization, the lack of replication in regional meta-analyses, varied eGFR transformation used in the pan African meta-analysis, high genetic differences among continental African populations and varied ascertainment procedures across the cohorts "

5. The coloc analyses with molecular traits are interesting. However, please comment on the ancestry of the molecular trait data (GTEx, NepthQTL) and discuss how ancestry mis-match might affect the results. Please specify how the LD reference data were handled in this context.

Response: Thank you for your comment. The GTEx V8 and NepthQTL (derived from participants of the NEPTUNE study¹ datasets are dominated by targets samples of recent European ancestry. These constraints the choice of the Colocalization: the select method is LD free with a single shared causal allele assumption between the target GWAS and eQTL traits. Given this limitation, we highlighted loci where there was evidence of violation of the assumption. Given the setup, it may be fruitful in subsequent work to investigate the application of multi-trait multi-group fine mapping ² methods for scenarios similar to the current one and/or leverage more recent larger and cosmopolitan eQTL GWAS ³ with suitable LD reference and methods. We have listed this as one of the limitation of this study on lines 482-485 reads " However, our study was limited by ancestry mismatch between our data and the GTEx data used for colocalization, the lack of replication in regional meta-analyses, varied eGFR transformation used in the pan African meta-analysis, high genetic differences among continental African populations and varied ascertainment procedures across the cohorts "

Reference

1. Han, S. K. *et al.* Mapping genomic regulation of kidney disease and traits through high-resolution and interpretable eQTLs. *Nat Commun* **14**, 2229 (2023).
2. Zhou, F. *et al.* Leveraging information between multiple population groups and traits improves fine-mapping resolution. *Nat Commun* **14**, 7279 (2023).
3. Orchard, P. *et al.* Cross-cohort analysis of expression and splicing quantitative trait loci in TOPMed. 2025.02.19.25322561 Preprint at <https://doi.org/10.1101/2025.02.19.25322561> (2025).

6. In the PGS analyses, I am a bit confused about what meta-analyses are used for what testing

set (ie, when were the pan-African-ancestry meta-analyses used, and when only the continental meta-analyses?). Please make this more clear.

Response: Thank you for your comment and suggestion. For polygenic scores development, we used all three regional meta-analyses (east, west, and south), we used pan-African meta-analysis, we further used summary data from MVP+UKBB+CKDgen and multi-ancestry meta-analysis of eGFR from previous study. On lines 268- 271 we wrote that “GWAS summary data from regional (east, west and south), continental, diaspora (MVP+UKBB+CKDgen), pan-African, and multi-ancestry meta-analyses were used to derive PGSs in MEIRU testing cohort in Malawi. We then compared the performances and transferability of the PGSs in MEIRU validation cohort”

7. “R2 is the difference between the fully adjusted and null model” should this not be reported as the the difference between the fully adjusted and null model divided by the residual variance, so $(R2_{full} - R2_{null})/(1-R2_{null})$?

Response: Thank you for your comment and suggestion. However, R2 is the incremental variance, and it is usually derived by the difference between the full model and null model. The full model includes PGS and other covariates, whereas the null model included only the covariates without PGS. We have also observed that the formula suggested by the reviewer tend to over inflate the R2 (see table below) and to keep it consistent with previous studies ^{1,2,3}, also use the current formul to derive R2 for PRSs.

$R2=(Full-Null)$	Full.R2	Null.R2	P	Number of SNP	Population	$R2=(Full-Null)/(1-null)$
0.1103	0.2168	0.2157	0.037	4959	South	0.1406
0.0002	0.2157	0.2157	0.916	28627	East	0.0003
0.0217	0.2160	0.2157	0.354	80030	West	0.0277
0.0121	0.2094	0.2093	0.491	1416184	Continental	0.0154
0.0217	0.2095	0.2093	0.357	782622	Pan-Africa	0.0275
0.0687	0.2100	0.2093	0.101	722	Diaspora	0.0869
0.0071	0.2158	0.2157	0.596	2133	Multi-ancestry	0.0090

References

- 1.Reay, W. R. et al. Using Genetics to Inform Interventions Related to Sodium and Potassium in Hypertension. *Circulation* 149, 1019–1032 (2024).
2. Örd, T. et al. Dissecting the polygenic basis of atherosclerosis via disease-associated cell state signatures. *Am J Hum Genet* 110, 722–740 (2023).
3. Sun, Q. et al. Improving polygenic risk prediction in admixed populations by explicitly modeling ancestral-differential effects via GAUDI. *Nat Commun* 15, 1016 (2024).

RESULTS

8. Please also briefly summarize what loci are not novel (to better highlight whether the stronger known loci are recapitulated in your African GWAS analysis)

Response: Thank you for the comments. In this revision, we have highlighted the stronger known loci that were significantly associated with eGFR. On lines 389-392 Interestingly, we found several loci which have been previous reported to be associated with eGFR (Table. 2). Of the previous loci, the most well established is the rs1145085 (GATM), previously reported to be as-sociated with eGFR, creatinine levels and CKD in individuals of European and Asian ancestries. GATM plays a crucial role in creatinine biosynthesis.

9. Please more clearly highlight what the inflation factors and LDSC intercepts were for each of the regional analyses.

Response: Thank you for your comments and suggestion, The inflation factors for regional meta-analysis were 1.013, 1.004, and 1.039, respectively for west, south, and east African meta-analysis see lines 297-299. We performed our regional meta-analyses using GWAMA. However, GWAMA does not compute the LDSC and we did not compute LDSC intercepts for the regional meta-analyses.

10. As mentioned in the major comments, the lack of replication across the African regions, as described by the authors, is rather worrisome. However, it is not clear whether the authors simply mean that the loci are not genome-wide significant in the other regions, or whether the loci show no association at all. This should be more clearly defined. Ie, what does the overlap look like when you take a liberal cutoff like $P < 1e-4$ for these loci?

Response: Thank you for your comment. Yes, we observed that the regional meta-analyses loci lacked replication among the three geographical regions. The lack of replication in this context means the loci are not statistically significant at $p < 0.05$ except for rs1706775 as shown in table and figure above in response to reviewer comment one. The lack of replication in other ancestry populations is due to the fact these loci are African specific (see Table.1), meaning they are monomorphic in individuals of European and Asian ancestry. Interestingly, our regional variants are noted to be significant (p -value < 0.05) in more than one of contributing studies to the regional meta-analyses as shown in figure above. However, there are two variants (rs115943222 and rs7763270) which were only found in one contributing study probably due to differences in the imputation densities and therefore could not be replicated in other contributing studies. These changes have been made on lines 303-307 and lines 394-409 of this revision.

11. The authors report relatively lower effect sizes of APOL1 variants on kidney function as compared to previous publications. Assuming the APOL1 variants are well-imputed, I think the authors should better discuss how phenotyping might affect the associations. Notably, it should be noted that the authors defined low kidney function as being under a certain eGFR threshold, whereas previous studies typically used EHR diagnoses as outcomes. Please discuss how this might impact the effect size estimation.

We appreciate the reviewer's comment and agree that differences in phenotyping may contribute to the variation in effect sizes observed across studies. In a recent study by Gbadegesin et al.,

the authors investigated the association between APOL1 variants and chronic kidney disease (CKD) in hospital-based cohorts from Nigeria and Ghana. Their definition of CKD included eGFR <90 mL/min/1.73 m², a urinary albumin-to-creatinine ratio ≥30 mg/g, or both. In addition, a proportion of their cases included individuals with biopsy-confirmed glomerular disease and those with sickle cell disease, a known risk factor for kidney impairment. This phenotyping, combined with a case-control design, can enhance study power to detect stronger effect sizes.

In contrast, our study used a cross-sectional, population-based approach relying on a single measurement of eGFR to define low eGFR (reduced kidney function). While eGFR is a useful indicator of renal function at the population level, it is not the same as a clinical diagnosis of CKD, especially in the absence of longitudinal data. Thus, our classification of kidney function status for APOL1 analysis in our study and study design may contribute to the lower effect sizes observed in our study relative to prior reports.

We have added this statement to the discussion section in lines 449-459 of this revision.

In the study of Gbadegesin et al., the authors reported an association between APOL1 variants and chronic kidney disease (CKD) in cohorts from Nigeria and Ghana. Their definition of CKD included eGFR <90 mL/min/1.73 m², a urinary albumin-to-creatinine ratio ≥30 mg/g, or both. In addition, a proportion of their cases included individuals with biopsy-confirmed glomerular disease and those with sickle cell disease, a known risk factor for kidney impairment. In contrast, our study used a cross-sectional, population-based approach relying on a single measurement of eGFR to define low eGFR. While eGFR is a useful indicator of renal function at the population level, it is not the same as a clinical diagnosis of CKD, especially in the absence of longitudinal data. Thus, our classification of kidney function status for APOL1 analysis, the demography of participants, and study design may contribute to the lower effect sizes observed in our study compared to previous findings. Detection of APOL1 with a stronger effect size is mostly seen in end-stage kidney disease, focal segmental glomerulosclerosis, and HIV-associated nephropathy studies.

12. In the results, it might be worth highlighting more prominently that the much larger multi-ancestry PGS performed worse than African-specific PGS. Also please discuss some of the R² values in the text, so the reader can better follow the story.

Response: Thank you for your comment and suggestion. In this revision we have included the results of R² values in the text and highlighted that the larger multi-ancestry derived PGS performed poorly compared to the southern Africa meta-analysis derived PGS. The predictive performance of PGSs were R²=0.11% with p-value=0.037 for southern Africa, R²=0.00028% with p-value=0.916 for eastern Africa, R²=0.022% with p-value=0.347 for western Africa, R²=0.012% with p-value=0.491 for continental Africa, R²=0.022% with p-value=0.357 for pan-Africa, R²=0.068% with p-value=0.101 for Africa, and R²=0.007% with p-value=0.596 for the multi-ancestry meta-analysis (Fig.3c). These revision has been made on lines 376-384

DISCUSSION

13. “In our analysis, we noted that rs6670659, rs74383679, and rs7763270 were common genetic variants in Africa” it is probably helpful to also highlight the nearby gene so the reader better understands the associated loci.

Response: Thank you for your comment and suggestion. We have revised the statement and now it reads: “In our analysis, we noted that rs6670659 (*HSD3BP3*), rs74383679 (*FAM72C*), and rs7763270 (*LAMA4*) were common genetic variants in Africa” on lines 413 and 414 of the revision.

14. Please also discuss how the various ascertainment procedures across the various cohorts might affect the GWAS results.

We thank the reviewer for this insightful question regarding the potential impact of ascertainment procedures on our GWAS results. We acknowledge that the diverse nature of the cohorts included in our pan-African meta-analysis, particularly the continent-specific cohorts, likely involved varying ascertainment strategies. These differences can indeed influence the genetic architecture observed and potentially affect the generalizability and power of our GWAS findings

In the discussion section, from line 481 to 488, we have included the following sentences. “The varied ascertainment procedures across the cohorts likely impacted your GWAS results by influencing sample representativeness, genetic diversity captured, and the observed genetic architecture of eGFR. For example, MVP cohort drawn from clinical settings may have enriched for individuals with lower eGFR or kidney disease, potentially increasing power for disease-related variants but overestimating effect sizes for the general population. Conversely, population-based cohorts would offer more generalized findings. Crucially, the diverse geographic origins of your African cohorts, alongside African American populations, meant sampling distinct ancestral genetic backgrounds, which may directly affected allele frequencies and the discovery of region-specific loci”

Reviewer #3 (Remarks to the Author):

This study presents a GWAS meta-analysis on estimated glomerular filtration rate (eGFR) across multiple African populations. The authors conduct regional, continental, and pan-African meta-analyses. While this study aims to address the underrepresentation of African populations in GWAS, I have major concerns regarding the robustness of the findings.

Major Concerns:

1. Lack of replication for novel variants. One of the main concerns is that the novel variants identified in the regional meta-analyses fail to replicate in other GWAS included in the study. In standard GWAS practice, replication in independent datasets is essential to validate findings and minimize false positives. The authors mention a "lack of overlapping genetic loci," but they do not provide specific association results, significance thresholds, or statistical replication power. Without these details, it is unclear whether the variants are nominally significant ($P < 0.05$) but fail to reach genome-wide significance, or if they show no signal at all.

Response: Thank you for your comment. Yes, we observed that the regional meta-analyses loci lacked replication among the three geographical regions. The lack of replication in this context means the loci are not statistically significant at $p < 0.05$ except for rs1706775 as shown in table and figure above in response to reviewer comment one. The lack of replication in other ancestry populations is due to the fact these loci are African specific (see Table.1 below in response to reviewer comment one), meaning they are monomorphic in individuals of European and Asian ancestry. Interestingly, our regional variants are noted to be significant (p -value < 0.05) in more than one of contributing studies to the regional meta-analyses as shown in figure below. However, there are two variants (rs115943222 and rs7763270) which were only found in one contributing study probably due to differences in the imputation densities and therefore could not be replicated in other contributing studies. These changes have been made on lines 303-307 and lines 394-409 of this revision.

Table 1. Minor allele frequency distribution of independent loci associated with eGFR in individuals of African ancestry in the regional meta-analysis

SNP	Genes	CHR	BP	EA	NEA	EAF	BETA	SE	P-value	Region	EAF- AFR	EAF- EUR	EAF- EAS
rs6670659	HSD3BP3	1	120090766	C	G	0.769	0.11	0.019	1.24E-08	Eastern	0.27	0	0
rs74383679*	FAM72C	1	206198189	G	C	0.155	0.159	0.028	2.75E-08	Southern	0.16	0	0
rs7763270*	LAMA4	6	112537967	A	G	0.139	-0.193	0.034	2.95E-08	Western	0.13	0	0
rs73788952*	OPRM1	6	154342741	G	A	0.092	0.132	0.022	5.04E-09	Southern	0.1	0	0
rs115943222*	KLHL1	13	70304786	A	C	0.009	-0.505	0.092	4.72E-08	Southern	0.03	0	0
rs1706775	SLC28A2	15	45586652	T	C	0.572	0.133	0.015	3.85E-17	Eastern	0.4	0	0
rs4243062*	LOC645752	15	78189084	T	C	0.299	-0.101	0.018	3.19E-08	Eastern	0.27	0.06	0.13

SNP; single nucleotide polymorphism, CHR chromosome, BP, base pair position, EA; effect allele, NEA, non-effect allele, EAF; Effect allele frequency, SE; standard error, * novel loci, AFR; African, EUR; European, EAS; East Asian ancestry

The fact that these novel regional variants do not emerge in the pan-African meta-analysis is particularly concerning. If these variants were truly region-specific, their absence of association elsewhere would be expected. However, upon examining the UK Biobank African ancestry subgroup, I found that all these variants exist at similar allele frequencies as reported in Table 1. This suggests that their absence in replication efforts is unlikely to be due to regional specificity. If these variants genuinely influence eGFR, they should be detectable in larger independent datasets with sufficient statistical power. The authors need to investigate and explain why these associations fail to replicate.

Response: Thank you for your comment and suggestion. We investigated why our regional variants are not replicated among the regional as well as other data like UKBB and MVP despite having almost same allele frequency distribution. What we found is that our regional variants successfully replicated within the region (see figure above) at p-value < 0.05. For instance, the UGR wave 1 study found rs4243062 to be associated with eGFR at p-value =0.047, this variant was replicated in UGR wave 2 study, AWI-Gen-Kenya study and AADM Kenya study with a p-value of 0.000221, 0.0046 and 0.00468, respectively as shown in figure below. However, the same variant (rs4243062) failed to replicate at p-value <0.05 in studies from western and southern Africa, suggesting the crucial role of regional environmental factors. Moreover, individuals of African ancestry in the UKBB are predominantly west African ancestry - and obviously have different environmental exposure which may influence this finding. These changes have been added on lines 394-409 of this revision.

To improve transparency, the authors should provide association results for all reported variants across each meta-analysis. Additionally, they should clarify whether their statements regarding "lack of overlap" refers to the lack of presence of these variants in different populations (which is doubtful, given their presence in the UK Biobank) or to differences in association results. Throughout the manuscript the authors do not say what significance thresholds they are using when discussing replication. For example, in line 379, when the authors state that "none was found to be significant in the regional and continental meta-analyses," it is unclear what significance threshold is being used.

Response: Thank you for your comment and suggestion. In this revision, we have provided all the associations results from our participating studies and cohorts as supplementary figure 2. The term lack of overlap was used for loci that are African specific and not found in other ancestry population. However, we have revised this statement now we call it lack replication at p-value < 0.05, on lines 303 and 304.

2. Lack of individual GWAS results in meta-analyses

The manuscript presents findings from meta-analyses but does not report results from the individual GWAS that contribute to each meta-analysis. This omission makes it impossible to determine whether the reported signals are robust across all datasets or driven by a single study. A well-conducted meta-analysis should demonstrate consistency of effect sizes across studies; otherwise, associations driven by only one dataset are more likely to be false positives.

Response: Thank you for your comment and suggestion. In this revision, we have included and presented the results of initial GWASs from participating cohort and studies. These results can be found in the first paragraph of the results section. Moreover, the Manhattan and the QQ plot of these GWASs are provided in supplementary figure 2.

To address this, the authors should provide forest plots or heterogeneity statistics to illustrate the consistency of effects across contributing GWAS. These analyses would clarify whether the reported associations are truly replicated across datasets or are disproportionately influenced by a single study. Without this information, it is difficult to assess the reliability of the reported associations.

Response: Thank you for the comments and suggestions. In this revision, we have provided the results of the initial GWASs that contributed to the regional meta-analysis. These results are included as supplementary material. We have further provided the forest plot of the results from contributing studies or cohort see figure above.

3. Claimed novel loci close to GATM

The manuscript identifies rs12595073 (SORD2P) and rs1918516 (SQRDL) as novel loci, but both variants are in close proximity to GATM locus, which is one of the strongest eGFR-associated signal in the genome. Given their location, these variants most likely reflect linkage disequilibrium with the well-established GATM association rather than representing independent signals. The authors present regional locus plots with a 500kb window, but a broader view would reveal that these loci are at the "tails" of the GATM signal. After conditioning on all independent GATM associated variants, it is doubtful that these variants would remain significant. The

authors identify independent loci by “clumping 500kb around lead SNPs” but in this instance the window needs to be extended further.

Response: Thank you for the comment and suggestions. Yes, we performed a clumping distance of 1000KB around the lead SNP to define a locus as suggested. The rs12595073 (SORD2P) and rs1918516 (SQRDL) falls with the clumping region. However, the LD among these loci reveal that these loci are in linkage equilibrium and are not tails of GATM loci see figure below. Moreover, we performed a conditional analysis using COJO method implemented in GCTA on GATM (rs1145085) using rs12595073, and rs1918516. We then checked how p-value of rs1145085 had been changed by condition on the two SNPs. The p-value PC of rs1145085 doesn't have a change for both conditional analyses (see table below) suggesting that these loci are independent of each other.

Conditional analysis on the GATM using COJO methods implemented in the GCTA														
Chr	SNP	bp	refA	freq	b	se	p	n	freq_gen	bC	bC_se	pC	diff_bp	bp_cond
15	15:45657804:A:G	45657804	A	0.2062	0.094	0.006	1.64E-53	80264	0.158	0.094	0.006	3.09E-53	504399	45153405
15	15:45657804:A:G	45657804	A	0.2062	0.094	0.006	1.64E-53	80264	0.158	0.093	0.006	1.08E-51	-503679	46161483

4. No comparison with European allele frequencies or GWAS results

The manuscript would benefit from comparison between the reported variants and European GWAS findings. Specifically, reporting European minor allele frequencies (MAFs) for the variants identified in the pan-African meta-analysis and providing effect size comparisons or scatter plots would help contextualize these findings. Understanding how these associations compare across ancestries could provide insights into potential genetic and environmental influences on eGFR. It is not clear which of the identified variants are African specific.

Response: Thank you for your comment and suggestion. In this revision, we have compared the effect allele frequency distribution of our loci with that of individuals of European and Asian ancestry. We found that three loci were monomorphic in the individuals of European and Asian ancestry as shown in table below

Table 2 Independent loci associated with eGFR in individuals of African ancestry in the pan-African meta-analysis.

SNP	Genes	Region	CHR	BP	NEA	EA	EAF	Zscore	P-value	EAF-EUR	EAF-EAS
rs11894953	TPRKB	Upstream	2	73964631	C	T	0.451	-5.546	2.92E-08	0.69	0.68
rs4859682	SHROOM3	Intronic	4	77410318	C	A	0.077	-5.499	3.81E-08	0.43	0.20
rs141647693*	ARG1	Intergenic	6	131810450	C	T	0.152	5.708	1.14E-08	0.01	0.00
rs316020	SLC22A2	Intronic	6	160669081	G	A	0.183	6.763	1.35E-11	0.11	0.04
rs13230509	UNCX	Intergenic	7	1286192	G	C	0.180	-5.654	1.56E-08	0.68	0.31
rs4236709	NRG1	Intronic	8	32410110	G	A	0.328	5.538	3.07E-08	0.81	0.78
rs7848018	C9orf3	Intronic	9	97748906	C	A	0.483	-6.125	9.08E-10	0.02	0.15
rs334	HBB	Exonic	11	5248232	T	A	0.099	-9.393	5.82E-21	0.00	0.00
rs77127179	CCKBR	Downstream	11	6293717	G	A	0.025	-5.551	2.85E-08	0.00	0.00
rs624307	SLC25A45	Exonic	11	65144075	C	T	0.101	6.519	7.07E-11	0.00	0.00
rs12595073*	SORD2P	ncRNA_intronic	15	45153405	G	T	0.102	-5.56	2.69E-08	0.04	0.39
rs1145085	GATM	Intronic	15	45657804	G	A	0.133	15.4	1.63E-53	0.72	0.18
rs1918516*	SQRDL	ncRNA_intronic	15	46161483	G	T	0.137	5.772	7.85E-09	0.71	0.96
rs8034430	UBE2Q2	Intronic	15	76169004	G	A	0.096	-7.502	6.30E-14	0.01	0.00
rs2453585	SLC47A1	Intronic	17	19447612	T	A	0.139	-6.225	4.82E-10	0.16	0.31
rs7208487	FBXL20	Intronic	17	37543449	G	T	0.341	-8.000	1.25E-15	0.16	0.24
rs9895661	BCAS3	ncRNA_intronic	17	59456589	C	T	0.488	6.294	3.10E-10	0.19	0.54
rs56376587	NFATC1	Intronic	18	77160235	C	A	0.151	5.632	1.78E-08	0.53	0.33
rs2096858	RNU6-1150P-201	Intergenic	21	45417467	T	C	0.016	6.361	2.00E-10	0.00	0.27

SNP; single nucleotide polymorphism, CHR chromosome, BP, base pair position, EA; effect allele, NEA, Non effect allele, EAF; Effect allele frequency, EUR; European. EAS; East Asian ancestry, * novel loci

Other points:

The abstract claims 28 loci and 5 novel variants from the regional meta-analysis. In the results text and Table 1 there are only 7 loci. Where are the other 21 loci?

Response: Thank you for your comment and suggestion. In this revision, we have corrected the error. Now the number of independent loci in regional meta-analyses is seven with five loci previously not reported to be associated with eGFR or CKD.

The abstract and results text claims 20 loci from the pan-African meta analysis, but Table 2 only has 19 variants.

Response: Thank you for your comment and suggestion. In this revision, we have corrected the error. Now the number of independent loci is 19 in both abstract and results section of the manuscript.

The authors claim 3 novel variants and point to Fig.S2 which has locus plots for four variants. I do not know why the fourth locus plot for 11:5513654 is there.

Response: Thank you're your comment: In this revision we have corrected the mistake. Now Fig.S2 has only three novel loci associated with eGFR in individuals of African ancestry.

Figures have small text that is very difficult to read.

Response: Thank you for your comment. In this revision, we have improved the resolution and increased the font size for our figures as shown below

The statement that the distribution of APOL1 variants 'demonstrates the diversity' of African populations might downplay the well-established genetic diversity of African populations. Reframing this to focus on the specific findings about APOL1 variants, within the broader context of known diversity, could improve clarity and avoid any potential for misunderstanding.

Response: We have reframed the sentence, and here's what we now have in the discussion on lines 431- 433 "The distribution of high-risk APOL1 variants for CKD is diverse across the Sub-Saharan African countries studied, in agreement with the substantial variations of APOL1 gene variants in African populations".

REVIEWER COMMENTS

Reviewer #1 (Remarks to the Author):

According to the summary form: “The genome-wide association study summary statistic data generated in this study have been deposited in the GWAS Catalog database under accession code [add hyperlink here].” Is the link available? Please include a Data Availability statement and accessions in the final manuscript.

Response: We thank the reviewer for their comments and suggestions. In this revision, we have provided the data statement which reads “Full summary statistics relating to the Million Veteran Program (MVP) studies are available at dbGAP accession phs001672.v2.p1. Individual-level data, phenotype, and genotype data of the continental Africa cohort are available to researchers under managed access on European Genome-phenome Archive (EGA): UGR: EGAS00001001558, EGAD00010000965, AWI-GEN: EGAD00001004448, and AADM: dbGAP: phs001844. Requests for access to data will be granted for all research consistent with the consent provided by participants” Moreover, this statement can be found on page 23, lines 569 to 575 of this revision.

Minor comments:

1. Make it clear in the Fig. 3c legend that the numbers on top of the bars are p-values.

Response: We thank the reviewer for this suggestion. We have now provided a figure legend which clearly states that the number above the bars on fig 3c are p-values. This revision is on page 35, lines 774–780, in the manuscript.

“Fig.3 PGSs results of eGFR in individuals of African ancestry. The MEIRU cohort was randomly divided into testing and validation cohorts. (a) Distribution of eGFR in the PGSs testing (n=3180) cohort (b) distribution of eGFR in the validation (n=3180) cohorts. (c) R² of PGSs in southern, eastern, western, continental, pan-African, and multi-ancestry meta-analyses, with their respective p-values on top of the bar plot. (d) Principal component analysis showing the genetic distance among the continental African cohorts.”

2. Please confirm the best R² PRS performance was R²=0.11%=0.0011 rather than R²=0.11. The manuscript conflicts with the response to reviewer 2, item 7.

Response: We thank the reviewer for their comment. We can confirm that the best performing PGSs was R²=0.11% as indicated in the manuscript on page 15, line 408.

Reviewer #2 (Remarks to the Author):

Nevertheless, a major issue regarding the consistency of the novel regional-specific loci remains. Notably, the authors argue that replication of these loci was not possible due to low allele frequencies in existing European and Asian ancestry datasets, which is a fair point. However, my comment did not pertain to replication within other major ancestries, but pertained to the consistency within the various regional African meta-analysis done by the authors. The regional-specific loci do not seem to replicate meaningfully in the other regional African GWAS. This

is an important point, as it could potentially mean that the regional-specific loci are not robust, and I think the authors need to be more transparent here. What were the allele frequencies of these loci in all regional meta-analyses? And what were the association results? If well-enough powered, one would expect some replication signal across African regions, unless the signals represent artifacts.

I should note that the authors do present a forest plot with – seemingly – the regional association results, although this figure does not seem to be referenced in the associated text and does not have a legend, making it difficult to understand. That being said, the weird patterns of association - for several of the loci – makes me concerned that harmonization of effect alleles was improperly performed in each sub-stratum prior to meta-analysis. For instance, it is very strange that 3 strata show significant positive effects for rs1706775, while two strata show nominally significant results in the opposite direction. The same seems to apply to rs6670659. On the one hand, this could explain the loss of significance in the larger meta-analysis (if the loci are actually bona fide), and on the other hand it should motivate a careful re-assessment of the harmonization steps taking prior to the meta-analyses.

Response: We thank the reviewer for their thoughtful and insightful comments. In response, we have carefully re-examined the seven previously reported regional lead loci and identified an error in the originally presented forest plots. Specifically, the forest plot data were based on effect estimates directly extracted from individual cohorts, which had not been fully harmonized, leading to inconsistencies in allele alignment.

To address this, we have revised the forest plots using harmonized summary statistics derived from our regional meta-analyses (**Table S6**), where we employed the GWAMA software. GWAMA automatically performs rigorous data harmonization, including allele alignment, prior to meta-analysis.

In this revised manuscript, we present only robust regional loci that meet the following stringent criteria.

- It reaches genome-wide significance ($P < 5 \times 10^{-8}$) in the regional meta-analysis and nominal significance ($P < 0.05$) in at least two of the contributing cohorts within that region.
- It shows a consistent direction of effect across all contributing cohorts in the region.

Based on this definition, we present revised Table S6 which reports the meta-analysis results from GWAMA including:

- Effect directions across contributing studies (which are now consistent),
- Meta-analysis effect sizes,
- Standard errors,
- Heterogeneity metrics including I^2 and Cochran's Q statistic

Here we report only loci that replicate within the same region. For example, rs73788952 locus is replicated in all different contributing countries in Southern region and rs6670659, rs1706775, rs4243062 replicated in all different contributing countries in Eastern region

Table. S6: The consistency of independent loci associated with eGFR from regional meta-analysis

SNP	EA	NEA	EAF	BETA	SE	p-value	Q statistic	Q p-value	I2	Studies	Samples	Effects	Region
rs6670659	C	G	0.769	0.110	0.019	1.24E-08	2.229	0.328	0.102	3	7765	+++?	East
rs1706775	T	C	0.572	0.133	0.015	3.85E-17	1.749	0.625	0.000	4	9834	++++	East
rs4243062*	T	C	0.299	-0.100	0.018	3.19E-08	3.027	0.387	0.008	4	9834	----	East
rs73788952*	G	A	0.092	0.132	0.022	5.04E-09	8.001	0.018	0.750	3	11687	+++	South

SNP; single nucleotide polymorphism, EA; effect allele, NEA; non-effect allele, EAF; effect allele frequency, SE; standard error, Studies; number of participating cohorts or studies from a region, Sample; number of individual from studies used in meta-analysis, +; BETA estimate was positively associated with eGFR, -; BETA estimate was negatively associated with eGFR, ?; SNPs not available in participating cohort or studies; *novel association

Figure. S3 Forest plot showing the effect size estimates (Beta) and 95% confidence intervals (CI) for the independent loci associated with eGFR. The horizontal lines represent 95% confidence intervals, and the black circles represent point estimates (Beta). P-values for each cohort are shown next to the respective estimates. The red dashed vertical line represents the null-effect line (Beta = 0). (a) **rs6670659** is an independent locus across three East African cohorts: Uganda Genome Resources (UGR) waves 1 and 2, and Africa Wits-INDEPTH partnership for Genomic Studies [AWI-Gen] East. (b) **rs1706775** is an independent locus across four East African cohorts: Uganda Genome Resources (UGR) waves 1 and 2, Africa America Diabetes (AADM) Kenya, and Africa Wits-INDEPTH partnership for Genomic Studies (AWI-Gen) East. (c) **rs4243062** is an independent novel locus across four East African cohorts: Uganda Genome Resources (UGR) waves 1 and 2, Africa America Diabetes (AADM), Kenya, and Africa Wits-INDEPTH partnership for Genomic Studies (AWI-Gen) East. (d) **rs73788952** is an independent novel locus across three Southern African cohorts: Malawi Epidemiology and Intervention Research Unit (MEIRU), African Research on Kidney Disease (ARK), and Africa Wits-INDEPTH partnership for Genomic Studies [AWI-Gen] South.

Table S4 Replication of independent loci among the three geographical regions in Africa

Meta-analysis eastern region								Meta-analysis southern region				Meta-analysis western region			
SNP	CHR	BP	EA	BETA	SE	P-value	Region	EA	BETA	SE	Pvalue	EA	BETA	SE	Pvalue
rs6670659	1	120090766	0.769	0.11	0.019	1.24E-08	Eastern	0.211	4.00E-03	1.60E-02	7.67E-01	0.269	8.00E-03	2.70E-02	7.60E-01
rs1706775	15	45586652	0.572	0.133	0.015	3.85E-17	Eastern	0.563	-2.70E-02	1.30E-02	4.00E-02	0.488	-1.80E-02	1.70E-02	2.74E-01
rs4243062	15	78189084	0.299	-0.101	0.018	3.19E-08	Eastern	0.742	6.00E-03	1.50E-02	6.89E-01	0.486	2.76E-02	1.70E-02	1.22E-01
Meta-analysis southern region								Meta-analysis Western region				Meta-analysis eastern region			
SNP	CHR	BP	EA	BETA	SE	P-value	Region	EA	BETA	SE	Pvalue	EA	BETA	SE	Pvalue
rs73788952	6	154342741	0.092	0.132	0.022	5.04E-09	Southern	0.087	-2.76E-02	4.23E-02	5.14E-01	0.929	-0.083	0.031	0.0093

The reviewer also indicated the need for replication across the African regions. We thank the reviewer for raising this concern, which has enabled us to re-examine these aspects and discuss their significance in the African context and the uniqueness of genetic findings in Africa.

African populations exhibit exceptionally high genetic diversity, and faster, more heterogeneous LD decay than Europeans and East Asians with greater variation within Africa than between Africans and Eurasians. This makes exact GWAS lead variants less likely to replicate uniformly across East, West, and Southern Africa even when the underlying causal loci are shared. As one review notes, “African populations are characterized by greater levels of genetic diversity, extensive population substructure, and less linkage disequilibrium (LD) among loci relative to non-Africans,” leading to shorter and more variable LD blocks that complicate direct marker-level replication across regions.¹⁻⁵

Regional LD patterns within Africa are not uniform, with documented differences between populations from West, East, and Southern Africa that affect tag SNP portability and imputation accuracy. For example, Tishkoff and colleagues reported “divergent patterns of LD” within Africa, including instances where “alleles that were in positive association in one population were in negative association in another,” and a resequencing study at IL13 found “divergent patterns of LD across West and East African populations,” underscoring region-specific haplotype structures that can impede exact SNP replication across regions⁵. Haplotype-based analyses further show substantial differences in LD extent and private haplotypes across sub-Saharan populations, directly impacting replication of index SNPs identified in one African cohort when tested in another.^{1,2,5}

Large Africa-wide resources demonstrate how regional structure, admixture, and unique haplotypes shape LD and GWAS transferability across the continent. The African Genome Variation Project reports “a substantial proportion of unshared (11%–23%) and novel (16%–24%) variants” across sampled populations, along with regionally distinct population structure and admixture.^{1,4}

Therefore, our results, which do not replicate across the region (See Table S4), align well with what has been reported in African populations as illustrated by the previous studies.

References

1. Gurdasani, D. *et al.* The African Genome Variation Project shapes medical genetics in Africa. *Nature* **517**, 327–332 (2015).
2. Campbell, M. C. & Tishkoff, S. A. AFRICAN GENETIC DIVERSITY: Implications for Human Demographic History, Modern Human Origins, and Complex Disease Mapping. *Annu Rev Genomics Hum Genet* **9**, 403–433 (2008).
3. Lambert, C. A. & Tishkoff, S. A. Genetic Structure in African Populations: Implications for Human Demographic History. *Cold Spring Harb Symp Quant Biol* **74**, 395–402 (2009).
4. Choudhury, A., Aron, S., Sengupta, D., Hazelhurst, S. & Ramsay, M. African genetic diversity provides novel insights into evolutionary history and local adaptations. *Hum Mol Genet* **27**, R209–R218 (2018).
5. Tarazona-Santos, E. & Tishkoff, S. A. Divergent patterns of linkage disequilibrium and haplotype structure across global populations at the interleukin-13 (IL13) locus. *Genes Immun* **6**, 53–65 (2005).

We have revised the abstract and manuscript on page 12, lines 305 to 322 of this revision and referenced the updated forest plot (figure S3 and Table S6 in the revised manuscript. We have also discussed the lack of external replication in the discussion section on page 17, lines 426 to 422 and wrote “ Our analysis, revealed that rs6670659 (*HSD3BP3*), rs73788952 (*OPRM1*), and rs1706775 (*SLC28A2*) were common genetic variants in Africa, yet monomorphic in European and Asian populations⁴⁶, underscoring the advantage of conducting genetic studies in individuals of African ancestry. We also found that our regional genetic loci were not replicated ($p < 0.05$) in region-based meta-analyses or in datasets such as UKB and MVP. African populations exhibit exceptionally high genetic diversity and faster, more heterogeneous LD decay than Europeans and East Asians, with greater variation within Africa than between Africans and Eurasians. This makes exact GWAS lead variants less likely to replicate uniformly across East, West, and Southern Africa even when the underlying causal loci are shared^{47–51}. Regional LD patterns within Africa are not uniform, with documented differences among populations from West, East, and Southern Africa that affect tag SNP portability and imputation accuracy. A previous study reported divergent patterns of LD within Africa, including instances where alleles that were in positive association in one population were in negative association in another, and a resequencing study at *IL13* found divergent patterns of LD across West and East African populations⁵¹, underscoring region-specific haplotype structures that can impede exact SNP replication across regions. Haplotype-based analyses further show substantial differences in LD extent and private haplotypes across sub-Saharan populations, directly impacting replication of index SNPs identified in one African cohort when tested in another^{48,51–53}. However, our loci were replicated ($p < 0.05$) with the same effect direction in more than one of contributing cohorts or studies to the regional meta-analyses. ”. These corrections strengthen the robustness of the regional associations and clarify the consistency and replication of the signals across the region.

Reviewer #3 (Remarks to the Author):

The newly provided tables and forest plots reinforce my concern that the seven regional lead SNPs are likely false positive associations rather than true, region-specific signals.

The forest plots for rs1706775 and rs6670659 reveal opposing effect directions across cohorts, yet the meta-analysis still produces highly significant associations, which does not make sense. For the other 5 variants, one dataset is mostly driving the association.

The authors refer to “replication within contributing studies,” but this is not replication in the GWAS sense. True replication requires observation of a consistent signal in an independent dataset or population. The critical issue is that within Africa, these signals do not replicate across regions or in independent African ancestry datasets (e.g., MVP or UK Biobank), despite similar allele frequencies. Without evidence of replication, either across regions or in other cohorts, and given the visible inconsistencies in the forest plots, there is no statistical or biological basis to present these loci as robust associations. Furthermore, no heterogeneity statistics are provided to assess consistency (e.g., I^2 , Q). At present, none of these seven loci meet the standard replication or robustness criteria expected of GWAS discoveries.

Response: We thank the reviewer for their thoughtful and insightful comments. In response, we have carefully re-examined the previously reported regional lead loci and identified an error in the originally presented forest plots. Specifically, the forest plot data were based on effect estimates directly extracted from individual cohorts, which had not been fully harmonized, leading to inconsistencies in allele alignment.

To address this, we have revised the forest plots using harmonized summary statistics derived from our regional meta-analyses, where we employed the GWAMA software. GWAMA automatically performs rigorous data harmonization, including allele alignment, prior to meta-analysis.

In this revised manuscript, we present only robust regional loci that meet the following stringent criteria.

- It reaches genome-wide significance ($P < 5 \times 10^{-8}$) in the regional meta-analysis and nominal significance ($P < 0.05$) in at least two of the contributing cohorts within that region.
- It shows a consistent direction of effect across all contributing cohorts in the region.

Based on this definition, we present revised Table S6 which reports the meta-analysis results from GWAMA including:

- Effect directions across contributing studies (which are now consistent),
- Meta-analysis effect sizes,
- Standard errors,
- Heterogeneity metrics including I^2 and Cochran’s Q statistics

Here we report only loci that replicate within the same region. For example, rs73788952 locus is replicated in all different contributing countries in Southern region and rs6670659, rs1706775, rs4243062 replicated in all different contributing countries in Eastern region

Table. S6: The consistency of independent loci associated with eGFR from regional meta-analysis

SNP	EA	NEA	EAF	BETA	SE	p-value	Q statistic	Q p-value	I2	Studies	Samples	Effects	Region
rs6670659	C	G	0.769	0.110	0.019	1.24E-08	2.229	0.328	0.102	3	7765	+++?	East
rs1706775	T	C	0.572	0.133	0.015	3.85E-17	1.749	0.625	0.000	4	9834	++++	East
rs4243062*	T	C	0.299	-0.100	0.018	3.19E-08	3.027	0.387	0.008	4	9834	----	East
rs73788952*	G	A	0.092	0.132	0.022	5.04E-09	8.001	0.018	0.750	3	11687	+++	South

SNP; single nucleotide polymorphism, EA; effect allele, NEA; non-effect allele, EAF; effect allele frequency, SE; standard error, Studies; number of participating cohorts or studies from a region, Sample; number of individual from studies used in meta-analysis, +; BETA estimate was positively associated with eGFR, -; BETA estimate was negatively associated with eGFR, ?; SNPs not available in participating cohort or studies; *novel association

Figure. S3 Forest plot showing the effect size estimates (Beta) and 95% confidence intervals (CI) for the independent loci associated with eGFR. The horizontal lines represent 95% confidence intervals, and the black circles represent point estimates (Beta). P-values for each cohort are shown next to the respective estimates. The red dashed vertical line represents the null-effect line (Beta = 0). (a) **rs6670659** is an independent locus across three East African cohorts: Uganda Genome Resources (UGR) waves 1 and 2, and Africa Wits-INDEPTH partnership for Genomic Studies [AWI-Gen] East. (b) **rs1706775** is an independent locus across four East African cohorts: Uganda Genome Resources (UGR) waves 1 and 2, Africa America Diabetes (AADM) Kenya, and Africa Wits-INDEPTH partnership for Genomic Studies (AWI-Gen) East. (c) **rs4243062** is an independent novel locus across four East African cohorts: Uganda Genome Resources (UGR) waves 1 and 2, Africa America Diabetes (AADM), Kenya, and Africa Wits-INDEPTH partnership for Genomic Studies (AWI-Gen) East. (d) **rs73788952** is an independent novel locus across three Southern African cohorts: Malawi Epidemiology and Intervention Research Unit (MEIRU), African Research on Kidney Disease (ARK), and Africa Wits-INDEPTH partnership for Genomic Studies [AWI-Gen] South.

Table S4 Replication of independent loci among the three geographical regions in Africa

Meta-analysis eastern region								Meta-analysis southern region				Meta-analysis western region			
SNP	CHR	BP	EA	BETA	SE	P-value	Region	EA	BETA	SE	Pvalue	EA	BETA	SE	Pvalue
rs6670659	1	120090766	0.769	0.11	0.019	1.24E-08	Eastern	0.211	4.00E-03	1.60E-02	7.67E-01	0.269	8.00E-03	2.70E-02	7.60E-01
rs1706775	15	45586652	0.572	0.133	0.015	3.85E-17	Eastern	0.563	-2.70E-02	1.30E-02	4.00E-02	0.488	-1.80E-02	1.70E-02	2.74E-01
rs4243062	15	78189084	0.299	-0.101	0.018	3.19E-08	Eastern	0.742	6.00E-03	1.50E-02	6.89E-01	0.486	2.76E-02	1.70E-02	1.22E-01
Meta-analysis southern region								Meta-analysis Western region				Meta-analysis eastern region			
SNP	CHR	BP	EA	BETA	SE	P-value	Region	EA	BETA	SE	Pvalue	EA	BETA	SE	Pvalue
rs73788952	6	154342741	0.092	0.132	0.022	5.04E-09	Southern	0.087	-2.76E-02	4.23E-02	5.14E-01	0.929	-0.083	0.031	0.0093

The reviewer also indicated the need for replication across the African regions. We thank the reviewer for raising this concern, which has enabled us to re-examine these aspects and discuss their significance in the African context and the uniqueness of genetic findings in Africa.

African populations exhibit exceptionally high genetic diversity, and faster, more heterogeneous LD decay than Europeans and East Asians with greater variation within Africa than between Africans and Eurasians. This makes exact GWAS lead variants less likely to replicate uniformly across East, West, and Southern Africa even when the underlying causal loci are shared. As one review notes, “African populations are characterized by greater levels of genetic diversity, extensive population substructure, and less linkage disequilibrium (LD) among loci relative to non-Africans,” leading to shorter and more variable LD blocks that complicate direct marker-level replication across regions. ¹⁻⁵

Regional LD patterns within Africa are not uniform, with documented differences between populations from West, East, and Southern Africa that affect tag SNP portability and imputation accuracy. For example, Tishkoff and colleagues reported “divergent patterns of LD” within Africa, including instances where “alleles that were in positive association in one population were in negative association in another,” and a resequencing study at IL13 found “divergent patterns of LD across West and East African populations,” underscoring region-specific haplotype structures that can impede exact SNP replication across regions ⁵. Haplotype-based analyses further show substantial differences in LD extent and private haplotypes across sub-Saharan populations, directly impacting replication of index SNPs identified in one African cohort when tested in another. ^{1,2,5}

Large Africa-wide resources demonstrate how regional structure, admixture, and unique haplotypes shape LD and GWAS transferability across the continent. The African Genome Variation Project reports “a substantial proportion of unshared (11%–23%) and novel (16%–24%) variants” across sampled populations, along with regionally distinct population structure and admixture. ^{1,4}

Therefore, our results, which do not replicate across the region (See Table S4), align well with what has been reported in African populations as illustrated by the previous studies.

References

- 1 Gurdasani, D. *et al.* The African Genome Variation Project shapes medical genetics in Africa. *Nature* **517**, 327–332 (2015).
- 2 Campbell, M. C. & Tishkoff, S. A. AFRICAN GENETIC DIVERSITY: Implications for Human Demographic History, Modern Human Origins, and Complex Disease Mapping. *Annu Rev Genomics Hum Genet* **9**, 403–433 (2008).
- 3 Lambert, C. A. & Tishkoff, S. A. Genetic Structure in African Populations: Implications for Human Demographic History. *Cold Spring Harb Symp Quant Biol* **74**, 395–402 (2009).
- 4 Choudhury, A., Aron, S., Sengupta, D., Hazelhurst, S. & Ramsay, M. African genetic diversity provides novel insights into evolutionary history and local adaptations. *Hum Mol Genet* **27**, R209–R218 (2018).
- 5 Tarazona-Santos, E. & Tishkoff, S. A. Divergent patterns of linkage disequilibrium and haplotype structure across global populations at the interleukin-13 (IL13) locus. *Genes Immun* **6**, 53–65 (2005).

We have revised the abstract and manuscript on page 12, lines 305 to 322 of this revision and referenced the updated forest plot (figure S3 and Table S6 in the revised manuscript. We have also discussed the lack of external replication in the discussion section on page 17, lines 426 to 422 and wrote “ Our analysis, revealed that rs6670659 (*HSD3BP3*), rs73788952 (*OPRM1*), and rs1706775 (*SLC28A2*) were common genetic variants in Africa, yet monomorphic in European and Asian populations⁴⁶, underscoring the advantage of conducting genetic studies in individuals of African ancestry. We also found that our regional genetic loci were not replicated ($p < 0.05$) in region-based meta-analyses or in datasets such as UKB and MVP. African populations exhibit exceptionally high genetic diversity and faster, more heterogeneous LD decay than Europeans and East Asians, with greater variation within Africa than between Africans and Eurasians. This makes exact GWAS lead variants less likely to replicate uniformly across East, West, and Southern Africa even when the underlying causal loci are shared^{47–51}. Regional LD patterns within Africa are not uniform, with documented differences among populations from West, East, and Southern Africa that affect tag SNP portability and imputation accuracy. A previous study reported divergent patterns of LD within Africa, including instances where alleles that were in positive association in one population were in negative association in another, and a resequencing study at *IL13* found divergent patterns of LD across West and East African populations⁵¹, underscoring region-specific haplotype structures that can impede exact SNP replication across regions. Haplotype-based analyses further show substantial differences in LD extent and private haplotypes across sub-Saharan populations, directly impacting replication of index SNPs identified in one African cohort when tested in another^{48,51–53}. However, our loci were replicated ($p < 0.05$) with the same effect direction in more than one of contributing cohorts or studies to the regional meta-analyses. ”. These corrections strengthen the robustness of the regional associations and clarify the consistency and replication of the signals across the region.

Additionally, two of the claimed novel loci (rs12595073 and rs1918516) are located immediately upstream and downstream of the well-established GATM locus. Given their proximity and the strength of the GATM signal, it is highly likely they reflect the extended association at this locus rather than independent signals. The appropriate analysis is to test whether these variants remain associated with eGFR after adjusting for the GATM variants. The author provided the conditional association the other way around. Without correct conditional analysis, these loci should not be presented as novel.

Response: We thank the reviewer for their thoughtful comment and suggestion. In this revision, we have assessed the novelty of two loci (rs12595073 and rs1918516) found near GATM. We have performed two conditional analyses using GCTA v1.94.4 with the --cojo-cond option. We conditioned the signal of rs12595073 on rs1145085 and rs1918516, and the results showed no substantial change in significance ($pC = 8.32e-08$ compared to $2.69e-08$). Furthermore, when we conditioned the signal of rs1918516 on rs12595073 and rs1145085, its significance was slightly reduced ($p = 7.85e-09$ vs. $pC = 7.68e-07$).

In this revision, we have included this statement “. Of the loci associated with eGFR in the pan-African ancestry meta-analysis, three were novel and had not previously been reported to be associated with eGFR or CKD. These loci included rs141647693 (*ARG1*), rs12595073 (*SORD2P*),

and rs1918516 (*SQRDL*) (**Figure S4**). However, we noted that rs12595073 and rs1918516 were located upstream and downstream of *GATM*, respectively. To validate the independence of these loci, we conditioned the signal of rs12595073 on rs1145085 and rs1918516. Conditional analyses were performed using the --cojo-cond option in the GCTA software (v1.94.4)⁴² with the AWI-Gen dataset as the genomic reference panel. The analysis parameters included a frequency difference threshold of 0.4 and a collinearity threshold of 0.9. Summary statistics were used, and genomic positions were extracted from a 5 Mb region surrounding *GATM*, including lead SNPs rs12595073, rs1918516, and rs141647693. Two independent conditional analyses were conducted: (i) rs12595073 was conditioned on rs1145085 and rs1918516, and (ii) rs1918516 was conditioned on rs1145085 and rs12595073. Our results showed no substantial change in significance ($p_C = 8.32e-08$ compared to $p = 2.69e-08$). Furthermore, when we conditioned the signal of rs1918516 on rs12595073 and rs1145085, its significance was slightly reduced ($p = 7.85e-09$ vs. $p_C = 7.68e-07$), suggesting that these loci are independent of each other, despite being located upstream and downstream of *GATM*” on page 13, lines 325 to 345 of this revision.